# Tight Certificates of Adversarial Robustness for Randomly Smoothed Classifiers

**Guang-He Lee**[1]**, Yang Yuan**[1,2]**, Shiyu Chang**[3]**, Tommi S. Jaakkola**[1]

[1]MIT Computer Science and Artificial Intelligence Lab
[2]Institute for Interdisciplinary Information Sciences, Tsinghua University
[3]MIT-IBM Watson AI Lab
{guanghe, yangyuan, tommi}@csail.mit.edu, shiyu.chang@ibm.com

## Abstract

Strong theoretical guarantees of robustness can be given for ensembles of classifiers generated by input randomization. Specifically, an $\ell_2$ bounded adversary cannot alter the ensemble prediction generated by an additive isotropic Gaussian noise, where the radius for the adversary depends on both the variance of the distribution as well as the ensemble margin at the point of interest. We build on and considerably expand this work across broad classes of distributions. In particular, we offer adversarial robustness guarantees and associated algorithms for the discrete case where the adversary is $\ell_0$ bounded. Moreover, we exemplify how the guarantees can be tightened with specific assumptions about the function class of the classifier such as a decision tree. We empirically illustrate these results with and without functional restrictions across image and molecule datasets.[1]

## 1 Introduction

Many powerful classifiers lack robustness in the sense that a slight, potentially unnoticeable manipulation of the input features, e.g., by an adversary, can cause the classifier to change its prediction [15]. The effect is clearly undesirable in decision critical applications. Indeed, a lot of recent work has gone into analyzing such failures together with providing certificates of robustness.

Robustness can be defined with respect to a variety of metrics that bound the magnitude or the type of adversarial manipulation. The most common approach to searching for violations is by finding an adversarial example within a small neighborhood of the example in question, e.g., using gradient-based algorithms [13, 15, 26]. The downside of such approaches is that failure to discover an adversarial example does not mean that another technique could not find one. For this reason, a recent line of work has instead focused on certificates of robustness, i.e., guarantees that ensure, for specific classes of methods, that no adversarial examples exist within a certified region. Unfortunately, obtaining exact guarantees can be computationally intractable [20, 25, 36], and guarantees that scale to realistic architectures have remained somewhat conservative [7, 27, 38, 39, 42].

Ensemble classifiers have recently been shown to yield strong guarantees of robustness [6]. The ensembles, in this case, are simply induced from randomly perturbing the input to a base classifier. The guarantees state that, given an additive isotropic Gaussian noise on the input example, an adversary cannot alter the prediction of the corresponding ensemble within an $\ell_2$ radius, where the radius depends on the noise variance as well as the ensemble margin at the given point [6].

In this work, we substantially extend robustness certificates for such noise-induced ensembles. We provide guarantees for alternative metrics and noise distributions (e.g., uniform), develop a stratified

likelihood ratio analysis that allows us to provide certificates of robustness over discrete spaces with respect to $\ell_0$ distance, which are *tight* and *applicable* to any measurable classifiers. We also introduce scalable algorithms for computing the certificates. The guarantees can be further tightened by introducing additional assumptions about the family of classifiers. We illustrate this in the context of ensembles derived from decision trees. Empirically, our ensemble classifiers yield the state-of-the-art certified guarantees with respect to $\ell_0$ bounded adversaries across image and molecule datasets in comparison to the previous methods adapted from continuous spaces.

## 2 Related Work

In a classification setting, the role of robustness certificates is to guarantee a constant classification within a local region; a certificate is always sufficient to claim robustness. When a certificate is both sufficient and necessary, it is called an exact certificate. For example, the exact $\ell_2$ certificate of a linear classifier is the $\ell_2$ distance between the classifier and a given point. Below we focus the discussions on the recent development of robustness guarantees for deep networks.

Most of the exact methods are derived on piecewise linear networks, defined as any network architectures with piecewise linear activation functions. Such class of networks has a mix integer-linear representation [22], which allows the usage of mix integer-linear programming [4, 9, 14, 25, 36] or satisfiability modulo theories [3, 12, 20, 33] to find the exact adversary under an $\ell_q$ radius. However, the exact method is in general NP-complete, and thus does not scale to large problems [36].

A certificate that only holds a sufficient condition is conservative but can be more scalable than exact methods. Such guarantees may be derived as a linear program [39, 40], a semidefinite program [30, 31], or a dual optimization problem [10, 11] through relaxation. Alternative approaches conduct layer-wise relaxations of feasible neuron values to derive the certificates [16, 27, 34, 38, 42]. Unfortunately, there is no empirical evidence of an effective certificate from the above methods in large scale problems. This does not entail that the certificates are not tight enough in practice; it might also be attributed to the fact that it is challenging to obtain a robust network in a large scale setting.

Recent works propose a new modeling scheme that ensembles a classifier by input randomization [2, 24], mostly done via an additive isotropic Gaussian noise. Lecuyer et al. [21] first propose a certificate based on differential privacy, which is improved by Li et al. [23] using Rényi divergence. Cohen et al. [6] proceed with the analysis by proving the *tight* certificate with respect to *all the measurable classifiers* based on the Neyman-Pearson Lemma [28], which yields the state-of-the-art provably robust classifier. However, the tight certificate is tailored to an isotropic Gaussian distribution and $\ell_2$ metric, while we generalize the result across broad classes of distributions and metrics. In addition, we show that such tight guarantee can be tightened with assumptions about the classifier.

Our method of certification also yields the first tight and actionable $\ell_0$ robustness certificates in discrete domains (cf. continuous domains where an adversary is easy to find [15]). Robustness guarantees in discrete domains are combinatorial in nature and thus challenging to obtain. Indeed, even for simple binary vectors, verifying robustness requires checking an exponential number of predictions for any black-box model.[2]

## 3 Certification Methodology

Given an input $\boldsymbol{x} \in \mathcal{X}$, a randomization scheme $\phi$ assigns a probability mass/density $\Pr(\phi(\boldsymbol{x}) = \boldsymbol{z})$ for each randomized outcome $\boldsymbol{z} \in \mathcal{X}$. We can define a probabilistic classifier either by specifying the associated conditional distribution $\mathbb{P}(y|\boldsymbol{x})$ for a class $y \in \mathcal{Y}$ or by viewing it as a random function $f(\boldsymbol{x})$ where the randomness in the output is independent for each $\boldsymbol{x}$. We compose the randomization scheme $\phi$ with a classifier $f$ to get a randomly smoothed classifier $\mathbb{E}_\phi[\mathbb{P}(y|\phi(\boldsymbol{x}))]$, where the probability for outputting a class $y \in \mathcal{Y}$ is denoted as $\Pr(f(\phi(\boldsymbol{x})) = y)$ and abbreviated as $p$, whenever $f, \phi, \boldsymbol{x}$ and $y$ are clear from the context. Under this setting, we first develop our framework for tight robustness certificates in §3.1, exemplify the framework in §3.2-3.4, and illustrate how the guarantees can be refined with further assumption in §3.5-3.6. We defer all the proofs to Appendix A.

## 3.1 A Framework for Tight Certificates of Robustness

In this section, we develop our framework for deriving tight certificates of robustness for randomly smoothed classifiers, which will be instantiated in the following sections.

**Point-wise Certificate.** Given $p$, we first identify a tight lower bound on the probability score $\Pr(f(\phi(\bar{\boldsymbol{x}})) = y)$ for another (neighboring) point $\bar{\boldsymbol{x}} \in \mathcal{X}$. Here we denote the set of measurable classifiers with respect to $\phi$ as $\mathcal{F}$. Without any additional assumptions on $f$, a lower bound can be found by the minimization problem:

$$\rho_{\boldsymbol{x},\bar{\boldsymbol{x}}}(p) \triangleq \min_{\bar{f} \in \mathcal{F}:\Pr(\bar{f}(\phi(\boldsymbol{x}))=y)=p} \Pr(\bar{f}(\phi(\bar{\boldsymbol{x}})) = y) \leq \Pr(f(\phi(\bar{\boldsymbol{x}})) = y). \tag{1}$$

Note that bound is tight since $f$ satisfies the constraint.

**Regional Certificate.** We can extend the point-wise certificate $\rho_{\boldsymbol{x},\bar{\boldsymbol{x}}}(p)$ to a regional certificate by examining the worst case $\bar{\boldsymbol{x}}$ over the neighboring region around $\boldsymbol{x}$. Formally, given an $\ell_q$ metric $\|\cdot\|_q$, the neighborhood around $\boldsymbol{x}$ with radius $r$ is defined as $\mathcal{B}_{r,q}(\boldsymbol{x}) \triangleq \{\bar{\boldsymbol{x}} \in \mathcal{X} : \|\boldsymbol{x} - \bar{\boldsymbol{x}}\|_q \leq r\}$. Assuming $p = \Pr(f(\phi(\boldsymbol{x})) = y) > 0.5$ for a $y \in \mathcal{Y}$, a robustness certificate on the $\ell_q$ radius can be found by

$$R(\boldsymbol{x}, p, q) \triangleq \sup r, \ s.t. \min_{\bar{\boldsymbol{x}} \in \mathcal{B}_{r,q}(\boldsymbol{x})} \rho_{\boldsymbol{x},\bar{\boldsymbol{x}}}(p) > 0.5. \tag{2}$$

Essentially, the certificate $R(\boldsymbol{x}, p, q)$ entails the following robustness guarantee:

$$\forall \bar{\boldsymbol{x}} \in \mathcal{X} : \|\boldsymbol{x} - \bar{\boldsymbol{x}}\|_q < R(\boldsymbol{x}, p, q), \text{ we have } \Pr(f(\phi(\bar{\boldsymbol{x}})) = y) > 0.5. \tag{3}$$

When the maximum can be attained in Eq. (2) (which will be the case in $\ell_0$ norm), the above $<$ can be replaced with $\leq$. Note that here we assume $\Pr(f(\phi(\boldsymbol{x})) = y) > 0.5$ and ignore the case that $0.5 \geq \Pr(f(\phi(\bar{\boldsymbol{x}})) = y) > \max_{y' \neq y} \Pr(f(\phi(\bar{\boldsymbol{x}})) = y')$. By definition, the certified radius $R(\boldsymbol{x}, p, q)$ is tight for binary classification, and provides a reasonable sufficient condition to guarantee robustness for $|\mathcal{Y}| > 2$. The tight guarantee for $|\mathcal{Y}| > 2$ will involve the maximum prediction probability over all the remaining classes (see Theorem 1 of [6]). However, when the prediction probability $p = \Pr(f(\phi(\boldsymbol{x})) = y)$ is intractable to compute and relies on statistical estimation for each class $y$ (e.g., when $f$ is a deep network), the tight guarantee is statistically challenging to obtain. The actual algorithm used by Cohen et al. [6] is also a special case of Eq. (2).

## 3.2 A Warm-up Example: the Uniform Distribution

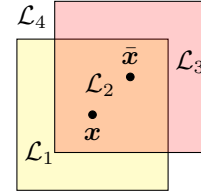

Figure 1: Uniform distributions.

To illustrate the framework, we show a simple (but new) scenario when $\mathcal{X} = \mathbb{R}^d$ and $\phi$ is an additive uniform noise with a parameter $\gamma \in \mathbb{R}_{>0}$:

$$\phi(\boldsymbol{x})_i = \boldsymbol{x}_i + \boldsymbol{\epsilon}_i, \boldsymbol{\epsilon}_i \overset{i.i.d.}{\sim} \text{Uniform}([-\gamma, \gamma]), \forall i \in \{1, \dots, d\}. \tag{4}$$

Given two points $\boldsymbol{x}$ and $\bar{\boldsymbol{x}}$, as illustrated in Fig. 1, we can partition the space $\mathbb{R}^d$ into 4 disjoint regions: $\mathcal{L}_1 = \mathcal{B}_{\gamma,\infty}(\boldsymbol{x}) \backslash \mathcal{B}_{\gamma,\infty}(\bar{\boldsymbol{x}})$, $\mathcal{L}_2 = \mathcal{B}_{\gamma,\infty}(\boldsymbol{x}) \cap \mathcal{B}_{\gamma,\infty}(\bar{\boldsymbol{x}})$, $\mathcal{L}_3 = \mathcal{B}_{\gamma,\infty}(\bar{\boldsymbol{x}}) \backslash \mathcal{B}_{\gamma,\infty}(\boldsymbol{x})$ and $\mathcal{L}_4 = \mathbb{R}^d \backslash (\mathcal{B}_{\gamma,\infty}(\bar{\boldsymbol{x}}) \cup \mathcal{B}_{\gamma,\infty}(\boldsymbol{x}))$. Accordingly, $\forall \bar{f} \in \mathcal{F}$, we can rewrite $\Pr(\bar{f}(\phi(\boldsymbol{x})) = y)$ and $\Pr(\bar{f}(\phi(\bar{\boldsymbol{x}})) = y)$ as follows:

$$\Pr(\bar{f}(\phi(\boldsymbol{x})) = y) = \sum_{i=1}^{4} \int_{\mathcal{L}_i} \Pr(\phi(\boldsymbol{x}) = \boldsymbol{z}) \Pr(\bar{f}(\boldsymbol{z}) = y)) d\boldsymbol{z} = \sum_{i=1}^{4} \pi_i \int_{\mathcal{L}_i} \Pr(\bar{f}(\boldsymbol{z}) = y)) d\boldsymbol{z},$$

$$\Pr(\bar{f}(\phi(\bar{\boldsymbol{x}})) = y) = \sum_{i=1}^{4} \int_{\mathcal{L}_i} \Pr(\phi(\bar{\boldsymbol{x}}) = \boldsymbol{z}) \Pr(\bar{f}(\boldsymbol{z}) = y)) d\boldsymbol{z} = \sum_{i=1}^{4} \bar{\pi}_i \int_{\mathcal{L}_i} \Pr(\bar{f}(\boldsymbol{z}) = y)) d\boldsymbol{z},$$

where $\pi_{1:4} = ((2\gamma)^{-d}, (2\gamma)^{-d}, 0, 0)$, and $\bar{\pi}_{1:4} = (0, (2\gamma)^{-d}, (2\gamma)^{-d}, 0)$. With this representation, it is clear that, in order to solve Eq. (1), we only have to consider the integral behavior of $\bar{f}$ within each region $\mathcal{L}_1, \dots, \mathcal{L}_4$. Concretely, we have:

$$\rho_{\boldsymbol{x},\bar{\boldsymbol{x}}}(p) = \min_{\bar{f} \in \mathcal{F}:\sum_{i=1}^{4} \pi_i \int_{\mathcal{L}_i} \Pr(\bar{f}(\boldsymbol{z})=y)) d\boldsymbol{z}=p} \sum_{i=1}^{4} \bar{\pi}_i \int_{\mathcal{L}_i} \Pr(\bar{f}(\boldsymbol{z}) = y)) d\boldsymbol{z}$$

$$= \min_{\substack{g:\{1,2,3,4\}\rightarrow[0,1], \\ \pi_1|\mathcal{L}_1|g(1)+\pi_2|\mathcal{L}_2|g(2)=p}} \bar{\pi}_2|\mathcal{L}_2|g(2) + \bar{\pi}_3|\mathcal{L}_3|g(3) = \min_{\substack{g:\{1,2,3,4\}\rightarrow[0,1], \\ \pi_1|\mathcal{L}_1|g(1)+\pi_2|\mathcal{L}_2|g(2)=p}} \bar{\pi}_2|\mathcal{L}_2|g(2),$$

where the second equality filters the components with $\pi_i = 0$ or $\bar{\pi}_i = 0$, and the last equality is due to the fact that $g(3)$ is unconstrained and minimizes the objective when $g(3) = 0$. Since $\pi_2 = \bar{\pi}_2$,

$$
\begin{cases}
\rho_{\boldsymbol{x},\bar{\boldsymbol{x}}}(p) = 0, & \text{if } 0 \leq p \leq \pi_1|\mathcal{L}_1| = \Pr(\phi(\boldsymbol{x}) \in \mathcal{L}_1), \\
\rho_{\boldsymbol{x},\bar{\boldsymbol{x}}}(p) = p - \pi_1|\mathcal{L}_1|, & \text{if } 1 \geq p > \pi_1|\mathcal{L}_1| = \Pr(\phi(\boldsymbol{x}) \in \mathcal{L}_1).
\end{cases}
$$

To obtain the regional certificate, the minimizers of $\min_{\bar{\boldsymbol{x}} \in \mathcal{B}_{r,q}(\boldsymbol{x})} \rho_{\boldsymbol{x},\bar{\boldsymbol{x}}}(p)$ are simply the points that maximize the volume of $\mathcal{L}_1 = \mathcal{B}_1 \setminus \mathcal{B}_2$. Accordingly,

**Proposition 1.** *If $\phi(\cdot)$ is defined as Eq. (4), we have $R(\boldsymbol{x}, p, q = 1) = 2p\gamma - \gamma$ and $R(\boldsymbol{x}, p, q = \infty) = 2\gamma - 2\gamma(1.5 - p)^{1/d}$.*

**Discussion.** Our goal here was to illustrate how certificates can be computed with the uniform distribution using our technique. However, the certificate radius itself is inadequate in this case. For example, $R(\boldsymbol{x}, p, q = 1) \leq \gamma$, which arises from the bounded support in the uniform distribution. The derivation nevertheless provides some insights about how one can compute the point-wise certificate $\rho_{\boldsymbol{x},\bar{\boldsymbol{x}}}(p)$. The key step is to partition the space into regions $\mathcal{L}_1, \ldots, \mathcal{L}_4$, where the likelihoods $\Pr(\phi(\boldsymbol{x}) = \boldsymbol{z})$ and $\Pr(\phi(\bar{\boldsymbol{x}}) = \boldsymbol{z})$ are both constant within each region $\mathcal{L}_i$. The property allows us to substantially reduce the optimization problem in Eq. (1) to finding a single probability value $g(i) \in [0, 1]$ for each region $\mathcal{L}_i$.

### 3.3 A General Lemma for Point-wise Certificate

In this section, we generalize the idea in §3.2 to find the point-wise certificate $\rho_{\boldsymbol{x},\bar{\boldsymbol{x}}}(p)$. For each point $\boldsymbol{z} \in \mathcal{X}$, we define the likelihood ratio $\eta_{\boldsymbol{x},\bar{\boldsymbol{x}}}(\boldsymbol{z}) \triangleq \Pr(\phi(\boldsymbol{x}) = \boldsymbol{z}) / \Pr(\phi(\bar{\boldsymbol{x}}) = \boldsymbol{z})$.[3] If we can partition $\mathcal{X}$ into $n$ regions $\mathcal{L}_1, \ldots, \mathcal{L}_n : \cup_{i=1}^n \mathcal{L}_i = \mathcal{X}$ for some $n \in \mathbb{Z}_{>0}$, such that the likelihood ratio within each region $\mathcal{L}_i$ is a constant $\eta_i \in [0, \infty]$: $\eta_{\boldsymbol{x},\bar{\boldsymbol{x}}}(\boldsymbol{z}) = \eta_i, \forall \boldsymbol{z} \in \mathcal{L}_i$, then we can sort the regions such that $\eta_1 \geq \eta_2 \geq \cdots \geq \eta_n$. Note that $\mathcal{X}$ can still be uncountable (see the example in §3.2).

Informally, we can always "normalize" $\bar{f}$ so that it predicts a constant probability value $g(i) \in [0, 1]$ within each likelihood ratio region $\mathcal{L}_i$. This preserves the integral over $\mathcal{L}_i$ and thus over $\mathcal{X}$, generalizing the scenario in §3.2. Moreover, to minimize $\Pr(\bar{f}(\phi(\bar{\boldsymbol{x}})) = y)$ under a fixed budget $\Pr(\bar{f}(\phi(\boldsymbol{x})) = y)$, as in Eq. (1), it is advantageous to set $\bar{f}(\boldsymbol{z})$ to $y$ in regions with high likelihood ratio. These arguments suggest a greedy algorithm for solving Eq. (1) by iteratively assigning $f(\boldsymbol{z}) = y, \forall \boldsymbol{z} \in \mathcal{L}_i$ for $i \in (1, 2, \ldots)$ until the budget constraint is met. Formally,

**Lemma 2.** $\forall \boldsymbol{x}, \bar{\boldsymbol{x}} \in \mathcal{X}$, $p \in [0, 1]$, *let* $H^* \triangleq \min_{H \in \{1, \ldots, n\}: \sum_{i=1}^H \Pr(\phi(\boldsymbol{x}) \in \mathcal{L}_i) \geq p} H$, *then* $\eta_{H^*} > 0$, *any* $f^*$ *satisfying Eq. (5) is a minimizer of Eq. (1),*

$$
\forall i \in \{1, 2, \ldots, n\}, \forall \boldsymbol{z} \in \mathcal{L}_i, \Pr(f^*(\boldsymbol{z}) = y) = \begin{cases}
1, & \text{if } i < H^*, \\
\frac{p - \sum_{i=1}^{H^*-1} \Pr(\phi(\boldsymbol{x}) \in \mathcal{L}_i)}{\Pr(\phi(\boldsymbol{x}) \in \mathcal{L}_{H^*})}, & \text{if } i = H^*, \quad (5) \\
0, & \text{if } i > H^*.
\end{cases}
$$

*and* $\rho_{\boldsymbol{x},\bar{\boldsymbol{x}}}(p) = \sum_{i=1}^{H^*-1} \Pr(\phi(\bar{\boldsymbol{x}}) \in \mathcal{L}_i) + (p - \sum_{i=1}^{H^*-1} \Pr(\phi(\boldsymbol{x}) \in \mathcal{L}_i))/\eta_{H^*}$

We remark that Eq. (1) and Lemma 2 can be interpreted as a likelihood ratio testing [28], by casting $\Pr(\phi(\boldsymbol{x}) = \boldsymbol{z})$ and $\Pr(\phi(\bar{\boldsymbol{x}}) = \boldsymbol{z})$ as likelihoods for two hypothesis with the significance level $p$. We refer the readers to [37] to see a similar Lemma derived under the language of hypothesis testing.

**Remark 3.** $\rho_{\boldsymbol{x},\bar{\boldsymbol{x}}}(p)$ *is an increasing continuous function of $p$; if $\eta_1 < \infty$, $\rho_{\boldsymbol{x},\bar{\boldsymbol{x}}}(p)$ is a strictly increasing continuous function of $p$; if $\eta_1 < \infty$ and $\eta_n > 0$, $\rho_{\boldsymbol{x},\bar{\boldsymbol{x}}} : [0, 1] \to [0, 1]$ is a bijection.*

Remark 3 will be used in §3.4 to derive an efficient algorithm to compute robustness certificates.

**Discussion.** Given $\mathcal{L}_i$, $\Pr(\phi(\boldsymbol{x}) \in \mathcal{L}_i)$, and $\Pr(\phi(\bar{\boldsymbol{x}}) \in \mathcal{L}_i), \forall i \in [n]$, Lemma 2 provides an $O(n)$ method to compute $\rho_{\boldsymbol{x},\bar{\boldsymbol{x}}}(p)$. For any actual randomization $\phi$, the key is to find a partition $\mathcal{L}_1, \ldots, \mathcal{L}_n$ such that $\Pr(\phi(\boldsymbol{x}) \in \mathcal{L}_i)$ and $\Pr(\phi(\bar{\boldsymbol{x}}) \in \mathcal{L}_i)$ are easy to compute. Having constant likelihoods in each $\mathcal{L}_i : \Pr(\phi(\boldsymbol{x}) = \boldsymbol{z}) = \Pr(\phi(\boldsymbol{x}) = \boldsymbol{z}'), \forall \boldsymbol{z}, \boldsymbol{z}' \in \mathcal{L}_i$ (cf. only having constant likelihood ratio $\eta_i$) is a way to simplify $\Pr(\phi(\boldsymbol{x}) \in \mathcal{L}_i) = |\mathcal{L}_i| \Pr(\phi(\boldsymbol{x}) = \boldsymbol{z})$, and similarly for $\Pr(\phi(\bar{\boldsymbol{x}}) \in \mathcal{L}_i)$.

### 3.4 A Discrete Distribution for $\ell_0$ Robustness

We consider $\ell_0$ robustness guarantees in a discrete space $\mathcal{X} = \left\{0, \frac{1}{K}, \frac{2}{K}, \ldots, 1\right\}^d$ for some $K \in \mathbb{Z}_{>0}$;[4] we define the following discrete distribution with a parameter $\alpha \in (0,1)$, independent and identically distributed for each dimension $i \in \{1, 2, \ldots, d\}$:

$$\begin{cases} \Pr(\phi(\boldsymbol{x})_i = \boldsymbol{x}_i) = \alpha, \\ \Pr(\phi(\boldsymbol{x})_i = z) = (1-\alpha)/K \triangleq \beta \in (0, 1/K), & \text{if } z \in \left\{0, \frac{1}{K}, \frac{2}{K}, \ldots, 1\right\} \text{ and } z \neq \boldsymbol{x}_i. \end{cases} \quad (6)$$

Here $\phi(\cdot)$ can be regarded as a composition of a Bernoulli random variable and a uniform random variable. Due to the symmetry of the randomization with respect to all the configurations of $\boldsymbol{x}, \bar{\boldsymbol{x}} \in \mathcal{X}$ such that $\|\boldsymbol{x} - \bar{\boldsymbol{x}}\|_0 = r$ (for some $r \in \mathbb{Z}_{\geq 0}$), we have the following Lemma for the equivalence of $\rho_{\boldsymbol{x}, \bar{\boldsymbol{x}}}$:

**Lemma 4.** *If $\phi(\cdot)$ is defined as Eq. (6), given $r \in \mathbb{Z}_{\geq 0}$, define the canonical vectors $\boldsymbol{x}_C \triangleq (0, 0, \cdots, 0)$ and $\bar{\boldsymbol{x}}_C \triangleq (1, 1, \cdots, 1, 0, 0, \cdots, 0)$, where $\|\bar{\boldsymbol{x}}_C\|_0 = r$. Let $\rho_r \triangleq \rho_{\boldsymbol{x}_C, \bar{\boldsymbol{x}}_C}$. Then for all $\boldsymbol{x}, \bar{\boldsymbol{x}}$ such that $\|\boldsymbol{x} - \bar{\boldsymbol{x}}\|_0 = r$, we have $\rho_{\boldsymbol{x}, \bar{\boldsymbol{x}}} = \rho_r$.*

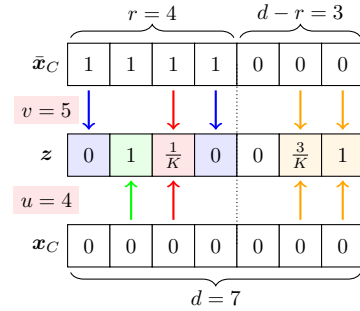

Figure 2: Illustration for Eq. (7)

Based on Lemma 4, finding $R(\boldsymbol{x}, p, q)$ for a given $p$, it suffices to find the maximum $r$ such that $\rho_r(p) > 0.5$. Since the likelihood ratio $\eta_{\boldsymbol{x}, \bar{\boldsymbol{x}}}(\boldsymbol{z})$ is always positive and finite, the inverse $\rho_r^{-1}$ exists (due to Remark 3), which allows us to pre-compute $\rho_r^{-1}(0.5)$ and check $p > \rho_r^{-1}(0.5)$ for each $r \in \mathbb{Z}_{\geq 0}$, instead of computing $\rho_r(p)$ for each given $p$ and $r$. Then $R(\boldsymbol{x}, p, q)$ is simply the maximum $r$ such that $p > \rho_r^{-1}(0.5)$. Below we discuss how to compute $\rho_r^{-1}(0.5)$ in a scalable way. Our first step is to identify a set of likelihood ratio regions $\mathcal{L}_1, \ldots, \mathcal{L}_n$ such that $\Pr(\phi(\boldsymbol{x}) \in \mathcal{L}_i)$ and $\Pr(\phi(\bar{\boldsymbol{x}}) \in \mathcal{L}_i)$ as used in Lemma 2 can be computed efficiently. Note that, due to Lemma 4, it suffices to consider $\boldsymbol{x}_C, \bar{\boldsymbol{x}}_C$ such that $\|\bar{\boldsymbol{x}}_C\|_0 = r$ throughout the derivation.

For an $\ell_0$ radius $r \in \mathbb{Z}_{\geq 0}, \forall (u, v) \in \{0, 1, \ldots, d\}^2$, we construct the region

$$\mathcal{L}(u, v; r) \triangleq \{\boldsymbol{z} \in \mathcal{X} : \Pr(\phi(\boldsymbol{x}_C) = \boldsymbol{z}) = \alpha^{d-u}\beta^u, \Pr(\phi(\bar{\boldsymbol{x}}_C) = \boldsymbol{z}) = \alpha^{d-v}\beta^v\}, \quad (7)$$

which contains points that can be obtained by "flipping" $u$ coordinates from $\boldsymbol{x}_C$ or $v$ coordinates from $\bar{\boldsymbol{x}}_C$. See Figure 2 for an illustration, where different colors represent different types of coordinates: orange means both $\boldsymbol{x}_C, \bar{\boldsymbol{x}}_C$ are flipped on this coordinate and they were initially the same; red means both are flipped and were initially different; green means only $\boldsymbol{x}_C$ is flipped and blue means only $\bar{\boldsymbol{x}}_C$ is flipped. By denoting the numbers of these coordinates as $i, j^*, u - i - j^*, v - i - j^*$, respectively, we have the following formula for computing the cardinality of each region $|\mathcal{L}(u, v; r)|$.

**Lemma 5.** *For any $u, v \in \{0, 1, \ldots, d\}, u \leq v, r \in \mathbb{Z}_{\geq 0}$ we have $|\mathcal{L}(u, v; r)| = |\mathcal{L}(v, u; r)|$, and*

$$|\mathcal{L}(u, v; r)| = \sum_{i=\max\{0, v-r\}}^{\min\left(u, d-r, \lfloor \frac{u+v-r}{2} \rfloor\right)} \frac{(K-1)^{j^*} r!}{(u-i-j^*)!(v-i-j^*)!j^*!} \frac{K^i (d-r)!}{(d-r-i)!i!},$$

*where $j^* \triangleq u + v - 2i - r$.*

Therefore, for a fixed $r$, the complexity of computing all the cardinalities $|\mathcal{L}(u, v; r)|$ is $\Theta(d^3)$. Since each region $\mathcal{L}(u, v; r)$ has a constant likelihood ratio $\alpha^{v-u}\beta^{u-v}$ and we have $\cup_{u=0}^d \cup_{v=0}^d \mathcal{L}(u, v; r) = \mathcal{X}$, we can apply the regions to find the function $\rho_{\boldsymbol{x}, \bar{\boldsymbol{x}}} = \rho_r$ via Lemma 2. Under this representation, the number of nonempty likelihood ratio regions $n$ is bounded by $(d+1)^2$, the perturbation probability $\Pr(\phi(\boldsymbol{x}) \in \mathcal{L}(u, v; r))$ used in Lemma 2 is simply $\alpha^{d-u}\beta^u |\mathcal{L}(u, v; r)|$, and similarly for the $\Pr(\phi(\bar{\boldsymbol{x}}) \in \mathcal{L}(u, v; r))$. Based on Lemma 2 and Lemma 5, we may use a for-loop to compute the bijection $\rho_r(\cdot)$ for the input $p$ until $\rho_r(p) = 0.5$, and return the corresponding $p$ as $\rho_r^{-1}(0.5)$. The procedure is illustrated in Algorithm 1.

**Scalable implementation.** In practice, Algorithm 1 can be challenging to implement; the probability values (e.g., $\alpha^{d-u}\beta^u$) can be extremely small, which is infeasible to be computationally represented using floating points. If we set $\alpha$ to be a rational number, both $\alpha$ and $\beta$ can be represented in fractions, and thus all the corresponding probability values can be represented by two (large) integers; we also observe that computing the (large) cardinality $|\mathcal{L}(u,v;r)|$ is feasible in modern large integer computation frameworks in practice (e.g., `python`), which motivates us to adapt the computation in Algorithm 1 to large integers.

For simplicity, we assume $\alpha = \alpha'/100$ with some $\alpha' \in \mathbb{Z} : 100 \geq \alpha' \geq 0$. If we define $\tilde{\alpha} \triangleq 100K\alpha \in \mathbb{Z}, \tilde{\beta} \triangleq 100K\beta \in \mathbb{Z}$, we may implement Algorithm 1 in terms of the non-normalized, integer version $\tilde{\alpha}, \tilde{\beta}$. Specifically, we replace $\alpha, \beta$ and the constant 0.5 with $\tilde{\alpha}, \tilde{\beta}$ and $50K \times (100K)^{d-1}$, respectively. Then all the computations in Algorithm 1 can be

---

**Algorithm 1** Computing $\rho_r^{-1}(0.5)$

1: sort $\{(u_i, v_i)\}_{i=1}^n$ by likelihood ratio
2: $p, \rho_r = 0, 0$
3: **for** $i = 1, \ldots, n$ **do**
4: $\quad p' = \alpha^{d-u_i}\beta^{u_i}$
5: $\quad \rho_r' = \alpha^{d-v_i}\beta^{v_i}$
6: $\quad \Delta\rho_r = \rho_r' \times |\mathcal{L}(u_i, v_i; r)|$
7: $\quad$ **if** $\rho_r + \Delta\rho_r < 0.5$ **then**
8: $\quad\quad \rho_r = \rho_r + \Delta\rho_r$
9: $\quad\quad p = p + p' \times |\mathcal{L}(u_i, v_i; r)|$
10: $\quad$ **else**
11: $\quad\quad p = p + p' \times (0.5 - \rho_r)/\rho_r'$
12: $\quad\quad$ return $p$
13: $\quad$ **end if**
14: **end for**

---

trivially adapted except the division $(0.5 - \rho_r)/\rho_r'$. Since the division is bounded by $|\mathcal{L}(u_i, v_i; r)|$ (see the comparison between line 9 and line 11), we can implement the division by a binary search over $\{1, 2 \ldots, |\mathcal{L}\{m_i, n_i\}|\}$, which will result in an upper bound with an error bounded by $\rho_r'$ in the original space, which is in turn bounded by $\alpha^d$ assuming $\alpha > \beta$. Finally, to map the computed, unnormalized $\rho_r^{-1}(0.5)$, denoted as $\tilde{\rho}_r^{-1}(0.5)$, back to the original space, we find an upper bound of $\rho_r^{-1}(0.5)$ up to the precision of $10^{-c}$ for some $c \in \mathbb{Z}_{>0}$ (we set $c = 20$ in the experiments): we find the smallest upper bound of $\tilde{\rho}_r^{-1}(0.5) \leq \hat{\rho} \times (10K)^c(100K)^{d-c}$ over $\hat{\rho} \in \{1, 2, \ldots, 10^c\}$ via binary search, and report an upper bound of $\rho_r^{-1}(0.5)$ as $\hat{\rho} \times 10^{-c}$ with an error bounded by $10^{-c} + \alpha^d$ in total. Note that an upper bound of $\rho_r^{-1}(0.5)$ is still a valid certificate.

As a side note, simply computing the probabilities in the log-domain will lead to uncontrollable approximate results due to floating point arithmetic; using large integers to ensure a verifiable approximation error in Algorithm 1 is necessary to ensure a computationally accurate certificate.

### 3.5 Connection Between the Discrete Distribution and an Isotropic Gaussian Distribution

When the inputs are binary vectors $\mathcal{X} = \{0, 1\}^d$, one may still apply the prior work [6] using an additive isotropic Gaussian noise $\phi$ to obtain an $\ell_0$ certificates since there is a bijection between $\ell_0$ and $\ell_2$ distance in $\{0, 1\}^d$. If one uses a denoising function $\zeta(\cdot)$ that projects each randomized coordinate $\phi(\boldsymbol{x})_i \in \mathbb{R}$ back to the space $\{0, 1\}$ using the (likelihood ratio testing) rule

$$\zeta(\phi(\boldsymbol{x}))_i = \mathbb{I}\{\phi(\boldsymbol{x})_i > 0.5\}, \forall i \in [d],$$

then the composition $\zeta \circ \phi$ is equivalent to our discrete randomization scheme with $\alpha = \Phi(0.5; \mu = 0, \sigma^2)$, where $\Phi$ is the CDF function of the Gaussian distribution with mean $\mu$ and variance $\sigma^2$.

If one applies a classifier upon the composition (or, equivalently, the discrete randomization scheme), then the certificates obtained via the discrete distribution is always tighter than the one via Gaussian distribution. Concretely, we denote $\mathcal{F}_\zeta \subset \mathcal{F}$ as the set of measurable functions with respect to the Gaussian distribution that can be written as the composition $\bar{f}' \circ \zeta$ for some $\bar{f}'$, and we have

$$\min_{\bar{f} \in \mathcal{F}_\zeta : \Pr(\bar{f}(\phi(\boldsymbol{x}))=y)=p} \Pr(\bar{f}(\phi(\bar{\boldsymbol{x}})) = y) \geq \min_{\bar{f} \in \mathcal{F} : \Pr(\bar{f}(\phi(\boldsymbol{x}))=y)=p} \Pr(\bar{f}(\phi(\bar{\boldsymbol{x}})) = y),$$

where the LHS corresponds to the certificate derived from the discrete distribution (i.e., applying $\zeta$ to an isotropic Gaussian), and the RHS corresponds to the certificate from the Gaussian distribution.

### 3.6 A Certificate with Additional Assumptions

In the previous analyses, we assume nothing but the measurability of the classifier. If we further make assumptions about the functional class of the classifier, we can obtain a tighter certificate than the ones outlined in §3.1. Assuming an extra denoising step in the classifier over an additive Gaussian noise as illustrated in §3.5 is one example.

Here we illustrate the idea with another example. We assume that the inputs are binary vectors $\mathcal{X} = \{0,1\}^d$, the outputs are binary $\mathcal{Y} = \{0,1\}$, and that the classifier is a decision tree that each input coordinate can be used at most once in the entire tree. Under the discrete randomization scheme, the prediction probability can be computed via tree recursion, since a decision tree over the discrete randomization scheme can be interpreted as assigning a probability of visiting the left child and the right child for each decision node. To elaborate, we denote $\mathtt{idx}[i]$, $\mathtt{left}[i]$, and $\mathtt{right}[i]$ as the split feature index, the left child and the right child of the $i^{\text{th}}$ node. Without loss of generality, we assume that each decision node $i$ routes its input to the right branch if $\boldsymbol{x}_{\mathtt{idx}[i]} = 1$. Then $\Pr(f(\phi(\boldsymbol{x})) = 1)$ can be found by the recursion

$$\mathtt{pred}[i] = \alpha^{\mathbb{I}\{\boldsymbol{x}_{\mathtt{idx}[i]} = 1\}}\beta^{\mathbb{I}\{\boldsymbol{x}_{\mathtt{idx}[i]} = 0\}}\mathtt{pred}[\mathtt{right}[i]] + \alpha^{\mathbb{I}\{\boldsymbol{x}_{\mathtt{idx}[i]} = 0\}}\beta^{\mathbb{I}\{\boldsymbol{x}_{\mathtt{idx}[i]} = 1\}}\mathtt{pred}[\mathtt{left}[i]], \quad (8)$$

where the boundary condition is the output of the leaf nodes. Effectively, we are recursively aggregating the partial solutions found in the left subtree and the right subtree rooted at each node $i$, and $\mathtt{pred}[\mathtt{root}]$ is the final prediction probability. Note that changing one input coordinate in $\boldsymbol{x}_k$ is equivalent to changing the recursion in the corresponding unique node $i'$ (if exists) that uses feature $k$ as the splitting index, which gives

$$\mathtt{pred}[i'] = \alpha^{\mathbb{I}\{\boldsymbol{x}_{\mathtt{idx}[i']} = 0\}}\beta^{\mathbb{I}\{\boldsymbol{x}_{\mathtt{idx}[i']} = 1\}}\mathtt{pred}[\mathtt{right}[i']] + \alpha^{\mathbb{I}\{\boldsymbol{x}_{\mathtt{idx}[i']} = 1\}}\beta^{\mathbb{I}\{\boldsymbol{x}_{\mathtt{idx}[i']} = 0\}}\mathtt{pred}[\mathtt{left}[i']].$$

In addition, changes in the left subtree do not affect the partial solution found in the right subtree, and vice versa. Hence, we may use dynamic programming to find the *exact* adversary under each $\ell_0$ radius $r$ by aggregating the worst case changes found in the left subtree and the right subtree rooted at each node $i$. See Appendix B.1 for details.

## 4 Learning and Prediction in Practice

Since we focus on the development of certificates, here we only briefly discuss how we train the classifiers and compute the prediction probability $\Pr(f(\phi(\boldsymbol{x})) = y)$ in practice.

**Deep networks**: We follow the approach proposed by the prior work [21]: training is conducted on samples drawn from the randomization scheme via a cross entropy loss. The prediction probability $\Pr(f(\phi(\boldsymbol{x})) = y)$ is estimated by the lower bound of the Clopper-Pearson Bernoulli confidence interval [5] with 100K samples drawn from the distribution and the $99.9\%$ confidence level. Since $\rho_{\boldsymbol{x},\bar{\boldsymbol{x}}}(p)$ is an increasing function of $p$ (Remark 3), a lower bound of $p$ entails a valid certificate.

**Decision trees**: we train the decision tree greedily in a breadth-first ordering with a depth limit; for each split, we only search coordinates that are not used before to enforce the functional constraint in §3.6, and optimize a weighted gini index, which weights each training example $\boldsymbol{x}$ by the probability that it is routed to the node by the discrete randomization. The details of the training algorithm is in Appendix B.2. The prediction probability is computed by Eq. (8).

## 5 Experiment

In this section, we validate the robustness certificates of the proposed discrete distribution ($\mathcal{D}$) in $\ell_0$ norm. We compare to the state-of-the-art additive isotropic Gaussian noise ($\mathcal{N}$) [6], since an $\ell_0$ certificate with radius $r$ in $\mathcal{X} = \{0, \frac{1}{K}, \dots, 1\}^d$ can be obtained from an $\ell_2$ certificate with radius $\sqrt{r}$. Note that the derived $\ell_0$ certificate from Gaussian distribution is still tight with respect to all the measurable classifiers (see Theorem 1 in [6]). We consider the following evaluation measures:

- $\mu(R)$: the average certified $\ell_0$ radius $R(\boldsymbol{x}, p, q)$ (with respect to the labels) across the testing set.
- ACC@$r$: the certified accuracy within a radius $r$ (the average $\mathbb{I}\{R(\boldsymbol{x}, p, q) \geq r\}$ in the testing set).

### 5.1 Binarized MNIST

We use a $55,000/5,000/10,000$ split of the MNIST dataset for training/validation/testing. For each data point $\boldsymbol{x}$ in the dataset, we binarize each coordinate by setting the threshold as $0.5$. Experiments are conducted on randomly smoothed CNN models and the implementation details are in Appendix C.1.

The results are shown in Table 1. For the same randomly smoothed CNN model (the $1^{\text{st}}$ and $2^{\text{nd}}$ rows in Table 1), our certificates are consistently better than the ones derived from the Gaussian

Table 1: Randomly smoothed CNN models on the MNIST dataset. The first two rows refer to the same model with certificates computed via different methods (see details in §3.5).

| $\phi$ | Certificate | $\mu(R)$ | ACC@$r$ | | | | | | |
|---|---|---|---|---|---|---|---|---|---|
| | | | $r=1$ | $r=2$ | $r=3$ | $r=4$ | $r=5$ | $r=6$ | $r=7$ |
| $\mathcal{D}$ | $\mathcal{D}$ | **3.456** | **0.921** | **0.774** | **0.539** | **0.524** | **0.357** | **0.202** | **0.097** |
| $\mathcal{D}$ | $\mathcal{N}$ [6] | 1.799 | 0.830 | 0.557 | 0.272 | 0.119 | 0.021 | 0.000 | 0.000 |
| $\mathcal{N}$ | $\mathcal{N}$ [6] | 2.378 | 0.884 | 0.701 | 0.464 | 0.252 | 0.078 | 0.000 | 0.000 |

Table 2: The guaranteed accuracy of randomly smoothed ResNet50 models on ImageNet.

| $\phi$ and certificate | ACC@$r$ | | | | | | |
|---|---|---|---|---|---|---|---|
| | $r=1$ | $r=2$ | $r=3$ | $r=4$ | $r=5$ | $r=6$ | $r=7$ |
| $\mathcal{D}$ | **0.538** | **0.394** | **0.338** | **0.274** | **0.234** | **0.190** | **0.176** |
| $\mathcal{N}$ [6] | 0.372 | 0.292 | 0.226 | 0.194 | 0.170 | 0.154 | 0.138 |

distribution (see §3.5). The gap between the average certified radius is about $1.7$ in $\ell_0$ distance, and the gap between the certified accuracy can be as large as $0.4$. Compared to the models trained with Gaussian noise (the 3$^{\text{rd}}$ row in Table 1), our model is also consistently better in terms of the measures.

Since the above comparison between our certificates and the Gaussian-based certificates is *relative*, we conduct an exhaustive search over all the possible adversary within $\ell_0$ radii 1 and 2 to study the tightness against the *exact* certificate. The resulting certified accuracies at radii 1 and 2 are $0.954$ and $0.926$, respectively, which suggest that our certificate is reasonably tight when $r = 1$ ($0.954$ vs. $0.921$), but still too pessimistic when $r = 2$ ($0.926$ vs. $0.774$). The phenomenon is expected since the certificate is based on *all the measurable functions* for the discrete distribution. A tighter certificate requires additional assumptions on the classifier such as the example in §3.6.

## 5.2 ImageNet

We conduct experiments on ImageNet [8], a large scale image dataset with $1,000$ labels. Following common practice, we consider the input space $\mathcal{X} = \{0, 1/255, \ldots, 1\}^{224 \times 224 \times 3}$ by scaling the images. We consider the same ResNet50 classifier [17] and learning procedure as Cohen et al. [6] with the only modification on the noise distribution. The details and visualizations can be found in Appendix C.2. For comparison, we report the best guaranteed accuracy of each method for each $\ell_0$ radius $r$ in Table 2. Our model outperforms the competitor by a large margin at $r = 1$ ($0.538$ vs. $0.372$), and consistently outperforms the baseline across different radii.

**Analysis.** We analyze our method in ImageNet in terms of 1) the number $n$ of nonempty likelihood ratio region $\mathcal{L}(u, v; r)$ in Algorithm 1, 2) the pre-computed $\rho_r^{-1}(0.5)$, and 3) the certified accuracy at each $\alpha$. The results are in Figure 3. For reproducability, the detailed accuracy numbers of 3) is available in Table 3 in Appendix C.2, and the pre-computed $\rho_r^{-1}(0.5)$ is available at our code repository. 1) The number $n$ of nonempty likelihood ratio regions is much smaller than the bound $(d+1)^2 = (3 \times 224 \times 224)^2$ for small radii. 2) The value $\rho_r^{-1}(0.5)$ approaches 1 more rapidly for a higher $\alpha$ value than a lower one. Note that $\rho_r^{-1}(0.5)$ only reaches 1 when $r = d$ due to Remark 3. Computing $\rho_r^{-1}(0.5)$ in large integer is time-consuming, which takes about 4 days for each $\alpha$ and $r$, but this can be trivially parallelized across different $\alpha$ and $r$.[5] For each radius $r$ and randomization parameter $\alpha$, note that the 4-day computation only has to be done *once*, and the pre-computed $\rho_r^{-1}(0.5)$ can be applied to any ImageNet scale images and models. 3) The certified accuracy behaves nonlinearly across different radii; relatively, a high $\alpha$ value exhibits a high certified accuracy at small radii and low certified accuracy at large radii, and vice versa.

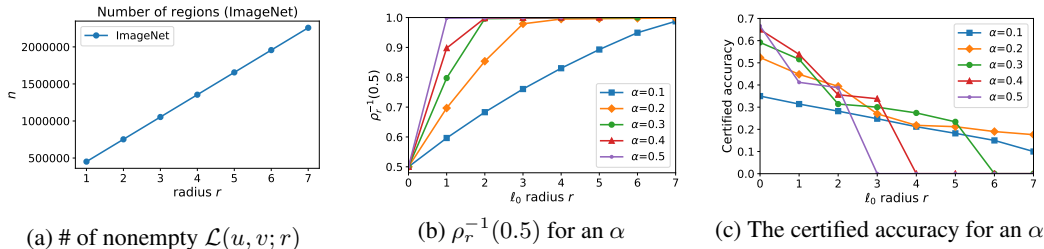

(a) # of nonempty $\mathcal{L}(u, v; r)$      (b) $\rho_r^{-1}(0.5)$ for an $\alpha$      (c) The certified accuracy for an $\alpha$

Figure 3: Analysis of the proposed method in the ImageNet dataset.

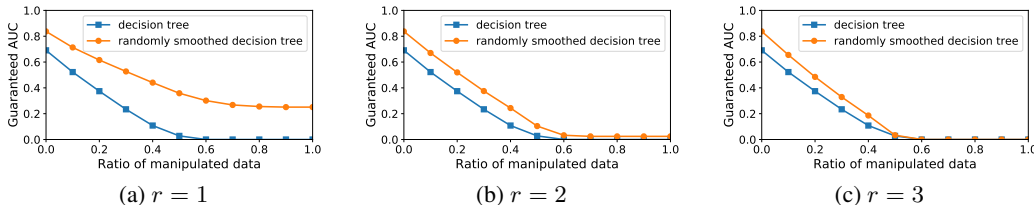

(a) $r = 1$              (b) $r = 2$              (c) $r = 3$

Figure 4: The guaranteed AUC in the Bace dataset across different $\ell_0$ radius $r$ and the ratio of testing data that the adversary can manipulate.

## 5.3 Chemical Property Prediction

The experiment is conducted on the Bace dataset [35], a binary classification dataset for biophysical property prediction on molecules. We use the Morgan fingerprints [32] to represent molecules, which are commonly used binary features [41] indicating the presence of various chemical substructures. The dimension of the features (fingerprints) is $1,024$. Here we focus on an ablation study comparing the proposed randomly smoothed decision tree with a vanilla decision tree, where the adversary is found by dynamic programming in §3.6 (thus the exact worse case) and a greedy search, respectively. More details can be found in Appendix C.3.

Since the chemical property prediction is typically evaluated via AUC [41], we define a robust version of AUC that takes account of the radius of the adversary as well as the ratio of testing data that can be manipulated. Note that to maximally decrease the score of AUC via a positive (negative) example, the adversary only has to maximally decrease (increase) its prediction probability, regardless of the scores of the other examples. Hence, given an $\ell_0$ radius $r$ and a ratio of testing data, we first compute the adversary for each testing data, and then find the combination of adversaries and the clean data under the ratio constraint that leads to the worst AUC score. See details in Appendix C.4.

The results are in Figure 4. Empirically, the adversary of the decision tree at $r = 1$ always changes the prediction probability of a positive (negative) example to $0$ ($1$). Hence, the plots of the decision tree model are constant across different $\ell_0$ radii. The randomly smoothed decision tree is consistently more robust than the vanilla decision tree model. We also compare the exact certificate of the prediction probability with the one derived from Lemma 2; the average difference across the training data is $0.358$ and $0.402$ when $r$ equals to $1$ and $2$, respectively. The phenomenon encourages the development of a classifier-aware guarantee that is tighter than the classifier-agnostic guarantee.

## 6 Conclusion

We present a stratified approach to certifying the robustness of randomly smoothed classifiers, where the robustness guarantees can be obtained in various resolutions and perspectives, ranging from a point-wise certificate to a regional certificate and from general results to specific examples. The hierarchical investigation opens up many avenues for future extensions at different levels.

## Acknowledgments

GH and TJ were in part supported by a grant from Siemens Corporation.

## Footnotes

[1]Project page: `http://people.csail.mit.edu/guanghe/randomized_smoothing`.

[2]We are aware of two concurrent works also yielding certificates in discrete domain [18, 19].

[3] If $\Pr(\phi(\bar{\boldsymbol{x}}) = \boldsymbol{z}) = \Pr(\phi(\boldsymbol{x}) = \boldsymbol{z}) = 0$, $\eta_{\boldsymbol{x},\bar{\boldsymbol{x}}}(\boldsymbol{z})$ can be defined arbitrarily in $[0, \infty]$ without affecting the solution in Lemma 2.

[4]More generally, the method applies to the $\ell_0$ / Hamming distance in a Hamming space (i.e., fixed length sequences of tokens from a discrete set, e.g., $(\spadesuit 10, \spadesuit J, \spadesuit Q, \spadesuit K, \spadesuit A) \in \{\spadesuit A, \spadesuit K, ..., \clubsuit 2\}^5$).

[5]As a side note, computing $\rho_r^{-1}(0.5)$ in MNIST takes less than 1 second for each $\alpha$ and $r$.

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
