[Supplementary Material · NeurIPS_robustness_supp.pdf]

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

# A   Proofs

To simplify exposition, we use $[n]$ to denote the set $\{1, 2, \ldots, n\}$.

## A.1   The proof of Proposition 1

*Proof.* We have

$$
\begin{cases}
\rho_{\boldsymbol{x}, \bar{\boldsymbol{x}}}(p) = 0, & \text{if } 0 \le p \le \Pr(\phi(\boldsymbol{x}) \in \mathcal{L}_1), \\
\rho_{\boldsymbol{x}, \bar{\boldsymbol{x}}}(p) = p - \Pr(\phi(\boldsymbol{x}) \in \mathcal{L}_1), & \text{if } 1 \ge p > \Pr(\phi(\boldsymbol{x}) \in \mathcal{L}_1),
\end{cases}
$$

where $\Pr(\phi(\boldsymbol{x}) \in \mathcal{L}_1) = \text{Vol}(\mathcal{B}_1 \backslash \mathcal{B}_2)/\text{Vol}(\mathcal{B}_1)$ and $\text{Vol}(\mathcal{B}_1)$ is a constant given $\gamma$. Hence, the minimizers of $\min_{\bar{\boldsymbol{x}} \in \mathcal{B}_{r,q}(\boldsymbol{x})} \rho_{\boldsymbol{x}, \bar{\boldsymbol{x}}}(p)$ are simply the points that maximize the volume of $\mathcal{B}_1 \backslash \mathcal{B}_2$, or, equivalently, minimize the volume of $\mathcal{B}_1 \cap \mathcal{B}_2$. Below we re-write $\bar{\boldsymbol{x}}$ as $\boldsymbol{x} + \boldsymbol{\delta}$.

*Case $q = 1$:* $\forall r > 0$, we want to find a $\boldsymbol{\delta}$ s.t., $\|\boldsymbol{\delta}\|_1 = r$ and the overlapping region is minimized: (By symmetry, we assume that $\boldsymbol{\delta}_i \ge 0$ for all $i$)

$$
\underset{\boldsymbol{\delta} \ge 0: \|\boldsymbol{\delta}\|_1 = r}{\arg\min} \prod_{i=1}^{d} (2\gamma - \boldsymbol{\delta}_i). \tag{9}
$$

Since $\forall i, j \in [d], i \ne j$, we know

$$
(2\gamma - \boldsymbol{\delta}_i)(2\gamma - \boldsymbol{\delta}_j) = 4\gamma^2 - (\boldsymbol{\delta}_i + \boldsymbol{\delta}_j) + \boldsymbol{\delta}_i \boldsymbol{\delta}_j \ge 4\gamma^2 - (\boldsymbol{\delta}_i + \boldsymbol{\delta}_j). \tag{10}
$$

So we can always move the mass of $\boldsymbol{\delta}_j$ to $\boldsymbol{\delta}_i$ to further decrease the product value. That means, $\boldsymbol{\delta}_1 = r, \boldsymbol{\delta}_i = 0, \forall i \ne 1$ minimizes Eq. (9) for a given $r$. As a result, we know

$$
\sup r, s.t. \min_{\boldsymbol{\delta}: \|\boldsymbol{\delta}\|_1 \le r} \rho_{\boldsymbol{x}, \boldsymbol{x}+\boldsymbol{\delta}}(p) > 0.5 \tag{11}
$$

$$
= \sup r, s.t. \ p - \left( 1 - \frac{(2\gamma)^{d-1}(2\gamma - r)}{(2\gamma)^d} \right) > 0.5 \tag{12}
$$

$$
= 2p\gamma - \gamma \tag{13}
$$

*Case $q = \infty$:* Similarly, for $q = \infty$ case, we want to find a $\boldsymbol{\delta}$ with $\|\boldsymbol{\delta}\|_\infty = r$, and the following is minimized: (by symmetry we assume $\boldsymbol{\delta}_i \ge 0$ for all $i$)

$$
\underset{\boldsymbol{\delta} \ge 0: \|\boldsymbol{\delta}\|_\infty = r}{\arg\min} \prod_{i=1}^{d} (2\gamma - \boldsymbol{\delta}_i).
$$

In this case, we should set $\boldsymbol{\delta}_i = r$ for all $i$, which means

$$
\sup r, s.t. \min_{\boldsymbol{\delta}: \|\boldsymbol{\delta}\|_\infty \le r} \rho_{\boldsymbol{x}, \boldsymbol{x}+\boldsymbol{\delta}}(p) > 0.5 \tag{14}
$$

$$
= \sup r, s.t. \ p - \left( 1 - \frac{(2\gamma - r)^d}{(2\gamma)^d} \right) > 0.5 \tag{15}
$$

$$
= \sup r, s.t. \ \frac{(2\gamma - r)^d}{(2\gamma)^d} > 1.5 - p \tag{16}
$$

It remains to see that

$$
\frac{(2\gamma - r)^d}{(2\gamma)^d} > 1.5 - p
$$

$$
\iff 2\gamma - r > 2\gamma(1.5 - p)^{1/d}
$$

$$
\iff 2\gamma - 2\gamma(1.5 - p)^{1/d} > r. \qquad \square
$$

## A.2 The proof of Lemma 2

*Proof.* $\forall \bar{f} \in \mathcal{F}$, We may rewrite the probabilities in an integral form:

$$\Pr(\bar{f}(\phi(\boldsymbol{x})) = y) = \sum_{i=1}^{n} \int_{\mathcal{L}_i} \Pr(\phi(\boldsymbol{x}) = \boldsymbol{z}) \Pr(\bar{f}(\boldsymbol{z}) = y) \mathrm{d}\boldsymbol{z},$$

$$\Pr(\bar{f}(\phi(\bar{\boldsymbol{x}})) = y) = \sum_{i=1}^{n} \int_{\mathcal{L}_i} \Pr(\phi(\bar{\boldsymbol{x}}) = \boldsymbol{z}) \Pr(\bar{f}(\boldsymbol{z}) = y) \mathrm{d}\boldsymbol{z}$$

Note that for all possible $\bar{f} \in \mathcal{F}$, we can re-assign all the function output within a likelihood region to be *constant* without affecting $\Pr(\bar{f}(\phi(\boldsymbol{x})) = y)$ and $\Pr(\bar{f}(\phi(\bar{\boldsymbol{x}})) = y)$. Concretely, we define $\bar{f}'$ as

$$\Pr(\bar{f}'(\boldsymbol{z}') = y) = \frac{\int_{\mathcal{L}_i} \Pr(\phi(\boldsymbol{x}) = \boldsymbol{z}) \Pr(\bar{f}(\boldsymbol{z}) = y) \mathrm{d}\boldsymbol{z}}{\int_{\mathcal{L}_i} \Pr(\phi(\boldsymbol{x}) = \boldsymbol{z}) \mathrm{d}\boldsymbol{z}}, \forall \boldsymbol{z}' \in \mathcal{L}_i, \forall i \in [n],$$

then we have

$$\int_{\mathcal{L}_i} \Pr(\phi(\boldsymbol{x}) = \boldsymbol{z}) \Pr(\bar{f}(\boldsymbol{z}) = y) \mathrm{d}\boldsymbol{z} = \int_{\mathcal{L}_i} \Pr(\phi(\boldsymbol{x}) = \boldsymbol{z}) \Pr(\bar{f}'(\boldsymbol{z}) = y) \mathrm{d}\boldsymbol{z}$$

Since in $\mathcal{L}_i$, $\Pr(\phi(\boldsymbol{x}) = \boldsymbol{z}) / \Pr(\phi(\bar{\boldsymbol{x}}) = \boldsymbol{z})$ is constant, we also have

$$\int_{\mathcal{L}_i} \Pr(\phi(\bar{\boldsymbol{x}}) = \boldsymbol{z}) \Pr(\bar{f}(\boldsymbol{z}) = y) \mathrm{d}\boldsymbol{z} = \int_{\mathcal{L}_i} \Pr(\phi(\bar{\boldsymbol{x}}) = \boldsymbol{z}) \Pr(\bar{f}'(\boldsymbol{z}) = y) \mathrm{d}\boldsymbol{z}$$

Therefore,

$$\Pr(\bar{f}(\phi(\boldsymbol{x})) = y) = \Pr(\bar{f}'(\phi(\boldsymbol{x})) = y), \text{ and}$$
$$\Pr(\bar{f}(\phi(\bar{\boldsymbol{x}})) = y) = \Pr(\bar{f}'(\phi(\bar{\boldsymbol{x}})) = y).$$

Hence, it suffices to consider the following program

$$(\mathrm{I}) \triangleq \min_{g:[n]\to[0,1]} \sum_{i=1}^{n} \int_{\mathcal{L}_i} \Pr(\phi(\bar{\boldsymbol{x}}) = \boldsymbol{z}) g(i) \mathrm{d}\boldsymbol{z},$$

$$s.t. \ \sum_{i=1}^{n} \int_{\mathcal{L}_i} \Pr(\phi(\boldsymbol{x}) = \boldsymbol{z}) g(i) \mathrm{d}\boldsymbol{z} = p,$$

where the optimum is equivalent to the program

$$\min_{\bar{f}\in\mathcal{F}:\Pr(\bar{f}(\phi(\boldsymbol{x}))=y)=p} \Pr(\bar{f}(\phi(\bar{\boldsymbol{x}})) = y),$$

and the each $g$ corresponds to a solution $\bar{f}$. For example, the $f^*$ in the statement corresponds to the $g^*$ defined as:

$$g^*(i) = \begin{cases} 1, & \text{if } i < H^*, \\ \frac{p - \sum_{i=1}^{H^*-1} \Pr(\phi(\boldsymbol{x})\in\mathcal{L}_i)}{\Pr(\phi(\boldsymbol{x})\in\mathcal{L}_{H^*})}, & \text{if } i = H^*, \\ 0, & \text{if } i > H^*. \end{cases} \tag{17}$$

We may simplify the program as

$$(\mathrm{I}) = \min_{g:[n]\to[0,1]} \sum_{i=1}^{n} \Pr(\phi(\bar{\boldsymbol{x}}) \in \mathcal{L}_i) g(i),$$

$$s.t. \ \sum_{i=1}^{n} \Pr(\phi(\boldsymbol{x}) \in \mathcal{L}_i) g(i) = p.$$

Clearly, if $\eta_i = 0$, all the optimal $g$ will assign $g(i) = 0$; our solution $g^*$ satisfies this property since

$$H^* \triangleq \min_{H\in\{1,\ldots,n\}:\sum_{i=1}^{H}\Pr(\phi(\boldsymbol{x})\in\mathcal{L}_i)\geq p} H \tag{18}$$

implies $\eta_{H^*} > 0$ (otherwise, it implies that $\Pr(\phi(\boldsymbol{x}) \in \mathcal{L}_{H^*}) = 0$ and leads to a contradiction). Hence, we can ignore the regions with $\eta_i = 0$, assume $\eta_n > 0$, and simplify program (I) again as

$$(\text{I}) = \min_{g:[n]\to[0,1]} \sum_{i=1}^{n} \frac{1}{\eta_i} \Pr(\phi(\boldsymbol{x}) \in \mathcal{L}_i) g(\eta_i), \tag{19}$$

$$s.t. \ \sum_{i=1}^{n} \Pr(\phi(\boldsymbol{x}) \in \mathcal{L}_i) g(\eta_i) = p. \tag{20}$$

It is evident that $g^*$ satisfies the constraint (20), and we will prove that any $g \neq g^*$ that satisfies constraint (20) cannot be better.

$\forall g : [n] \to [0, 1]$, we define $\Delta(i) \triangleq (g^*(i) - g(i)) P(\phi(\boldsymbol{x}) \in \mathcal{L}_i)$. Then we have

$$\sum_{i=1}^{n} \frac{1}{\eta_i} \Pr(\phi(\boldsymbol{x}) \in \mathcal{L}_i) g(i) = \sum_{i=1}^{n} \frac{1}{\eta_i} \left[ \Pr(\phi(\boldsymbol{x}) \in \mathcal{L}_i) g^*(i) - \Delta(i) \right]$$

$$= \sum_{i=1}^{H^*} \frac{1}{\eta_i} \Pr(\phi(\boldsymbol{x}) \in \mathcal{L}_i) g^*(i) - \sum_{i=1}^{n} \frac{1}{\eta_i} \Delta(i). \tag{21}$$

Note that $\Delta(i) \geq 0$ for $i < H^*$, $\Delta(i) \leq 0$ for $i > H^*$, and $\sum_{i=1}^{n} \Delta(\eta_i) = 0$ due to the constraint (20). Therefore, we have

$$\sum_{i=1}^{n} \frac{1}{\eta_i} \Delta(\eta_i) \leq \sum_{i=1}^{n} \frac{1}{\eta_{H^*}} \Delta(\eta_i) = 0. \tag{22}$$

Finally, combining (21) and (22),

$$\sum_{i=1}^{n} \frac{1}{\eta_i} \Pr(\phi(\boldsymbol{x}) \in \mathcal{L}_i) g(i) \geq \sum_{i=1}^{H^*} \frac{1}{\eta_i} \Pr(\phi(\boldsymbol{x}) \in \mathcal{L}_i) g^*(i). \tag{23}$$

$\square$

### A.3 The proof of Lemma 4

*Proof.* If $\varphi(\cdot)$ is defined as Eq. (6), $\forall \boldsymbol{x}, \bar{\boldsymbol{x}} \in \mathcal{X}$ such that $\|\boldsymbol{x} - \bar{\boldsymbol{x}}\|_0 = r$, below we show that $\rho_{\boldsymbol{x}, \bar{\boldsymbol{x}}}$ is independent of $\boldsymbol{x}$ and $\bar{\boldsymbol{x}}$. Indeed, since $\|\boldsymbol{x} - \bar{\boldsymbol{x}}\|_0$ is the number of non-zero elements of $\boldsymbol{x} - \bar{\boldsymbol{x}}$, we know there are exactly $r$ dimensions such that $\boldsymbol{x}$ and $\bar{\boldsymbol{x}}$ do not match. Notice that $\varphi(\cdot)$ applies to each dimension of $\boldsymbol{x}$ independently, so we can safely ignore any correlations between two dimensions. Therefore, by the symmetry of the distribution, we can rearrange the *order* of coordinates, and assume $\boldsymbol{x}$ and $\bar{\boldsymbol{x}}$ differ for the first $r$ dimensions, and match for the rest $d - r$ dimensions.

Notice that the randomization $\varphi(\cdot)$ has the nice property that the perturbing probabilities are oblivious to the *actual values* of the input. Therefore, by the definition of $\boldsymbol{x}_C, \bar{\boldsymbol{x}}_C$, we know that they are the canonical form of all pairs of $\boldsymbol{x}$ and $\bar{\boldsymbol{x}}$ such that $\|\boldsymbol{x} - \bar{\boldsymbol{x}}\|_0 = r$; hence, $\rho_{\boldsymbol{x}, \bar{\boldsymbol{x}}}(p)$ is constant and equals $\rho_r(p)$ for every $p \in [0, 1]$. $\square$

### A.4 The proof of Lemma 5

*Proof.* In this proof, we adopt the notation of the canonical form $\boldsymbol{x}_C$ and $\bar{\boldsymbol{x}}_C$ from Appendix A.3.

Recall that $\boldsymbol{x}_C$ is a zero vector, and $\bar{\boldsymbol{x}}_C$ has the first $r$ entries equal to 1 and the last $d - r$ entries equal to 0. We use the likelihood tuple $(u, v)$ to refer the scenario when $\boldsymbol{x}_C$ "flips" $u$ coordinates (the likelihood is $\alpha^{d-u}\beta^u$), and $\bar{\boldsymbol{x}}_C$ "flips" $v$ coordinates (the likelihood is $\alpha^{d-v}\beta^v$). Note that $u \leq v$ by assumption. For $r \in [d]$ and $(u, v) \in \{0, 1, \dots, d\}^2$, the number of possible outcome $\boldsymbol{z} \in \mathcal{X}$ with the likelihood tuple $(u, v)$ can be computed in the following way:

$$|\mathcal{L}(u, v; r)| = \sum_{i=0}^{\min(u,d-r)} \sum_{j=0}^{u-i} \frac{(K-1)^j \mathbb{I}((u-i-j) + (v-i-j) + j = r) r!}{(u-i-j)!(v-i-j)!j!} \frac{K^i (d-r)!}{(d-r-i)!i!},$$

where the first summation and the term

$$\frac{K^i(d-r)!}{(d-r-i)!i!}$$

correspond to the case where $i$ entries out of the last $(d-r)$ coordinates in $\boldsymbol{x}_C$ and $\bar{\boldsymbol{x}}_C$ are both modified. Notice that $\boldsymbol{x}_C$ and $\bar{\boldsymbol{x}}_C$ are equal in the last $d-r$ dimensions, so if $\boldsymbol{x}_C$ has $i$ entries modified among them, in order to ensure that $\bar{\boldsymbol{x}}_C$ equals $\boldsymbol{x}_C$ after modification, $\bar{\boldsymbol{x}}_C$ should have exactly the same $i$ entries modified as well (in order to become the same $\boldsymbol{z}$ in Eq. (7)).

The second summation and the term

$$\frac{(K-1)^j \mathbb{I}((u-i-j)+(v-i-j)+j=r)r!}{(u-i-j)!(v-i-j)!j!}$$

corresponds to the case where $j$ entries out of the first $r$ coordinates of $\boldsymbol{x}_C$ and $\bar{\boldsymbol{x}}_C$ are modified to any values other than $\{0,1\}$, $u-i-j$ entries in $\boldsymbol{x}_C$ are modified to 1, and $v-i-j$ entries in $\bar{\boldsymbol{x}}_C$ are modified to 0. By the same analysis, we know that both $\boldsymbol{x}_C$ and $\bar{\boldsymbol{x}}_C$ should have exactly the same $j$ entries modified to any value other than $\{0,1\}$. The indicator function $\mathbb{I}((u-i-j)+(v-i-j)+j=r)$ simply verifies that whether the value of $j$ is valid. Note that these two summations have covered all possible cases of modifications on $\boldsymbol{x}_C$ and $\bar{\boldsymbol{x}}_C$ in $\mathcal{L}(u,v;r)$.

After fixing the value of $i$ and $j$, each summand is simply calculating the number of symmetric cases. $(K-1)^j$ means there are $j$ entries modified to $(K-1)$ possible values. $\frac{r!}{(u-i-j)!(v-i-j)!j!}$ is the number of possible configurations for the first $r$ coordinates. $K^i$ means there are $i$ entries momdified to $K$ possible values. $\frac{(d-r)!}{(d-r-1)!i!}$ is the number of possible configurations for the last $d-r$ coordinates.

Now it remains to simplify the expression. Let $j^* \triangleq u+v-2i-r$, we have

$$
\begin{aligned}
|\mathcal{L}(u,v;r)| &= \sum_{i=0}^{\min(u,d-r)} \frac{\mathbb{I}(j^* \geq 0)\mathbb{I}(j^* \leq u-i)(K-1)^{j^*}r!}{(u-i-j^*)!(v-i-j^*)!j^*!} \frac{K^i(d-r)!}{(d-r-i)!i!}, \\
&= \sum_{i=\max\{0,v-r\}}^{\min(u,d-r)} \frac{\mathbb{I}(j^* \geq 0)(K-1)^{j^*}r!}{(u-i-j^*)!(v-i-j^*)!j^*!} \frac{K^i(d-r)!}{(d-r-i)!i!}, \\
&= \sum_{i=\max\{0,v-r\}}^{\min(u,d-r,\lfloor \frac{u+v-r}{2} \rfloor)} \frac{(K-1)^{j^*}r!}{(u-i-j^*)!(v-i-j^*)!j^*!} \frac{K^i(d-r)!}{(d-r-i)!i!}.
\end{aligned}
\tag{24}
$$

Moreover, we know that $|\mathcal{L}(u,v;r)| = |\mathcal{L}(v,u;r)|$ holds by the symmetry between $\boldsymbol{x}_C$ and $\bar{\boldsymbol{x}}_C$. □

# B    Algorithms For Decision Tree

## B.1    Dynamic Programming For Restricted Decision Tree

Given an input $\boldsymbol{x} \in \mathcal{X}$, we run dynamic programming (Algorithm 2) for computing the certificate, based on the same idea mentioned in Section 3.6.

**Algorithm 2** DP($\boldsymbol{x}$, $i$, $R$)

---

1: **if** $i$ is leaf **then**
2:     **for** $r = 1, \cdots, R$ **do**
3:         $\mathtt{adv}[i, r] =$Leaf-Output($i$)
4:     **end for**
5:     Return
6: **end if**
7: $rw = \alpha^{\mathbb{I}\{\boldsymbol{x}_{\mathtt{idx}[i]}=1\}} \beta^{\mathbb{I}\{\boldsymbol{x}_{\mathtt{idx}[i]}=0\}}$
8: $lw = \alpha^{\mathbb{I}\{\boldsymbol{x}_{\mathtt{idx}[i]}=0\}} \beta^{\mathbb{I}\{\boldsymbol{x}_{\mathtt{idx}[i]}=1\}}$
9: DP($\boldsymbol{x}$, $\mathtt{right[i]}$, $R$)
10: DP($\boldsymbol{x}$, $\mathtt{left[i]}$, $R$)
11: **for** $r = 1, \cdots, R$ **do**
12:     $\mathtt{adv}[i, r] = 1$
13:     **for** $\bar{r} = 0, \cdots r$ **do**
14:         $\mathtt{adv}[i, r] = \min\{\mathtt{adv}[i, r], rw * \mathtt{adv}[\mathtt{right}[i], \bar{r}] + lw * \mathtt{adv}[\mathtt{left}[i], r - \bar{r}]\}$
15:     **end for**
16:     **for** $\bar{r} = 0, \cdots r - 1$ **do**
17:         $\mathtt{adv}[i, r] = \min\{\mathtt{adv}[i, r], lw * \mathtt{adv}[\mathtt{right}[i], \bar{r}] + rw * \mathtt{adv}[\mathtt{left}[i], r - 1 - \bar{r}]\}$
18:     **end for**
19: **end for**

---

We use $\mathtt{adv}[i, r]$ to denote the worst prediction at node $i$ if at most $r$ features can be perturbed. Algorithm 2 uses the following updating rule for $\mathtt{adv}[i, r]$.

$$\mathtt{adv}[i, r] = \min\{$$
$$\min_{\bar{r} \in \{0,1,...,r\}} \{\alpha^{\mathbb{I}\{\boldsymbol{x}_{\mathtt{idx}[i]}=1\}} \beta^{\mathbb{I}\{\boldsymbol{x}_{\mathtt{idx}[i]}=0\}} \mathtt{adv}[\mathtt{right}[i], \bar{r}] + \alpha^{\mathbb{I}\{\boldsymbol{x}_{\mathtt{idx}[i]}=0\}} \beta^{\mathbb{I}\{\boldsymbol{x}_{\mathtt{idx}[i]}=1\}} \mathtt{adv}[\mathtt{left}[i], r - \bar{r}]\},$$
$$\min_{\bar{r} \in \{0,1,...,r-1\}} \{\alpha^{\mathbb{I}\{\boldsymbol{x}_{\mathtt{idx}[i]}=0\}} \beta^{\mathbb{I}\{\boldsymbol{x}_{\mathtt{idx}[i]}=1\}} \mathtt{adv}[\mathtt{right}[i], \bar{r}] + \alpha^{\mathbb{I}\{\boldsymbol{x}_{\mathtt{idx}[i]}=1\}} \beta^{\mathbb{I}\{\boldsymbol{x}_{\mathtt{idx}[i]}=0\}} \mathtt{adv}[\mathtt{left}[i], r - 1 - \bar{r}]\}\}$$

There are two cases in this updating rule. In the first case, the feature used at node $i$ is not perturbed, so it remains to see if we perturb $\bar{r}$ features in the right subtree and $r - \bar{r}$ features in the left subtree, what is the minimum adversarial prediction if $\bar{r} \in \{0, \cdots, r\}$. In the second case, the feature used at node $i$ is perturbed, and we check if we perturb $r - 1$ features in the two subtrees, what is the minimum adversarial prediction. Combining the two cases together, we get the solution for $\mathtt{adv}[i, r]$.

## B.2 Training Algorithm For Decision Tree

We consider the randomization scheme introduced in Eq. (6): for every coordinate in a given input $\boldsymbol{x}$, we may perturb its value with probability $\beta$. After perturbation, $\boldsymbol{x}$ may arrive at any leaf node, rather than following one specific path as in the standard decision tree. Therefore, when training, we maintain the probability of arriving at the current tree node for every input $\boldsymbol{x}$, denoted as $\mathtt{probs}$. The probability is multiplied by $\alpha$ or $\beta$ after each layer, depending on the input and the feature used for the current tree node. See Algorithm 3 for details.

The overall framework of Algorithm 3 is standard: we train the tree nodes greedily in breadth-first ordering, and pick the best splitting feature every time. However, when picking the best splitting feature, the standard decision tree uses Gini impurity based on all the remaining training data that will follow the path from the root to the current tree node. In our algorithm, this will include all the training data, but with different arriving probabilities. Therefore, we apply the weighted Gini impurity metric instead. Specifically, for a split, its weighted Gini impurity is (after $\mathtt{probs}$ is updated with $\mathtt{idx}$):

$$1 - \left(\frac{\sum_{\boldsymbol{x} \in \mathcal{X}, y=1} \mathtt{probs}[\boldsymbol{x}]}{\sum_{\boldsymbol{x} \in \mathcal{X}} \mathtt{probs}[\boldsymbol{x}]}\right)^2 - \left(\frac{\sum_{\boldsymbol{x} \in \mathcal{X}, y=0} \mathtt{probs}[\boldsymbol{x}]}{\sum_{\boldsymbol{x} \in \mathcal{X}} \mathtt{probs}[\boldsymbol{x}]}\right)^2$$

When the arriving probability for each $\boldsymbol{x}$ is restricted to be either 0 or 1, this definition becomes the standard Gini impurity.

**Algorithm 3** Train($X$,$Y$, `maxdep`)
---
1: $Q = [(\texttt{root}, 0, [1, 1, \cdots, 1])]$
2: **while** Q not Empty **do**
3:     $i, \texttt{dep}, \texttt{probs} = Q.\text{pop}()$
4:     **if** `dep=maxdep` **then**
5:         Assign-Leaf-Node($i$, `probs`)
6:         Continue
7:     **end if**
8:     f-list = Get-available-features()
9:     **for** `idx` in f-list **do**
10:         $\texttt{list} = [(y, \alpha^{\mathbb{I}\{\boldsymbol{x}_{\texttt{idx}}=1\}}\beta^{\mathbb{I}\{\boldsymbol{x}_{\texttt{idx}}=0\}}\texttt{probs}[\boldsymbol{x}])$ for $(\boldsymbol{x}, y)$ in $X]$
11:         $\texttt{idx}^* = $ Update-best-feature-score $(\texttt{idx}, \texttt{list}, \texttt{idx}^*)$
12:     **end for**
13:     $\texttt{idx}[i] = \texttt{idx}^*$
14:     `left-probs=probs`
15:     `right-probs=probs`
16:     **for** $\boldsymbol{x}$ in $X$ **do**
17:         **if** $\boldsymbol{x}_{\texttt{idx}} = 1$ **then**
18:             `left-probs`$[\boldsymbol{x}] = $`left-probs`$[\boldsymbol{x}] * \alpha$
19:             `right-probs`$[\boldsymbol{x}] = $`right-probs`$[\boldsymbol{x}] * \beta$
20:         **else**
21:             `right-probs`$[\boldsymbol{x}] = $`right-probs`$[\boldsymbol{x}] * \alpha$
22:             `left-probs`$[\boldsymbol{x}] = $`left-probs`$[\boldsymbol{x}] * \beta$
23:         **end if**
24:     **end for**
25:     $Q.\text{push}(\texttt{left}[i], \texttt{dep} + 1, \texttt{left-probs})$
26:     $Q.\text{push}(\texttt{right}[i], \texttt{dep} + 1, \texttt{right-probs})$
27: **end while**
---

## C   Experimental Details

### C.1   Supplementary Materials for the MNIST Experiment

For the MNIST experiment, we use a simple 4-layer convolutional network, where the first two layers are convolutional layers, and the last two layers are feedforward layers. For the two convolutional layers, we use kernel size 5, and output channels 20 and 50, respectively. For the two feedforward layers, we use 500 hidden nodes. We tune the hyperparameter $\alpha \in \{0.72, 0.76, 0.80, 0.84, 0.88, 0.92, 0.96\}$ for $\mu(R)$ in the validation set using the certificates from the Gaussian distribution [6]. The resulting $\alpha$ is 0.8. We also train a CNN with an isotropic Gaussian with the $\sigma$ that corresponds to $\alpha = 0.8$ (see §3.5).

The learning procedure for all the models are the same (except the distribution). The batch size is 400. We train each model for 30 epochs with the SGD optimizer with Nesterov momentum (momentum = 0.9). The learning rate is initially set to be 0.05 and annealed by a factor of 10 for every 10 epochs of training. The models are implemented in `PyTorch` [29], and run on single GPU with 12G memory.

### C.2   Supplementary Materials for the ImageNet Experiment

We use the `PyTorch` [29] implementation provided by Cohen et al. [6] as the backbone and implement our algorithm based on their pipeline. Thus, the training details are consistent to the one reported in their paper except that we use a different distribution. Here, we summarize some important details. ResNet-50 is used as the base classifier for our ImageNet experiment, whose architecture is provided in `torchvision`. After the randomization is done, we normalize each image by subtracting the dataset mean (0.485, 0.456, 0.406) and dividing by the standard deviation (0.229, 0.224, 0.225). Parameters are optimized by SGD with momentum set as 0.9. The learning rate is initially set to be 0.1 and annealed by a factor of 10 for every 30 epochs of training. The total number of training

Table 3: The guaranteed accuracy for different $\alpha$ of ResNet50 models smoothed by the discrete distribution on ImageNet.

| $\alpha$ value | ACC@$r$ | | | | | | | |
|---|---|---|---|---|---|---|---|---|
| | $r=0$ | $r=1$ | $r=2$ | $r=3$ | $r=4$ | $r=5$ | $r=6$ | $r=7$ |
| 0.5 | 0.666 | 0.412 | 0.388 | 0.000 | 0.000 | 0.000 | 0.000 | 0.000 |
| 0.4 | 0.650 | 0.538 | 0.356 | 0.338 | 0.000 | 0.000 | 0.000 | 0.000 |
| 0.3 | 0.592 | 0.516 | 0.314 | 0.300 | 0.274 | 0.234 | 0.000 | 0.000 |
| 0.2 | 0.524 | 0.448 | 0.394 | 0.270 | 0.218 | 0.212 | 0.190 | 0.176 |
| 0.1 | 0.350 | 0.314 | 0.282 | 0.248 | 0.212 | 0.182 | 0.150 | 0.100 |

Figure 5: ImageNet images corrupted by varying levels of the discrete noise.

epochs is 90. The batch size is 300, parallelized across 2 GPUs. We tune $\alpha \in \{0.1, 0.2, 0.3, 0.4, 0.5\}$ for the discrete distribution, and measure the performance in ACC@$r$, compared to the classifier under an additive isotropic Gaussian noise [6].[6] The samples of the randomized image for each $\alpha$ are visualized in Figure 5. We follow the prior work [6] to evaluate every 100th image in the validation set. The detailed accuracy numbers of our approach under different $\alpha$ and $r$ are available in Table 3.

### C.3 Supplementary Materials for the Chemical Property Prediction Experiment

The dataset contains $1,513$ molecules (data points). We split the data into the training, validation, and testing sets with the ratio 0.8, 0,1, and 0,1, respectively. Following common practice in chemical property prediction [41], the splitting is done based on the Bemis-Murcko scaffold [1]; the molecules within a split are inside different scaffolds from the other splits. We refer the details of scaffold splitting to [41]. We observe similar experiment results when we use a random split.

For both the decision tree and the randomly smoothed decision tree, we tune the depth limit in $\{6, 7, 8, 9, 10\}$. For the randomly smoothed decision tree, we also tune $\alpha \in \{0.7, 0.75, 0.8, 0.85, 0.9\}$. The tuned $\alpha$ is 0.8, and the tuned depth limits are 10 for both models.

### C.4 Computing Adversarial AUC

Assume that there are $n + m$ data points, $n$ of them are positive instances, denoted as $A \triangleq \{x_1, \cdots, x_n\}$, and $m$ of them are negative instances, denoted $B \triangleq \{x_{n+1}, \cdots, x_{n+m}\}$. Denote the whole dataset as $X \triangleq A \cup B$. For data point $x \in X$, we may adversarially perturb $x$ up to the perturbation radius $r$, denoted as $x^r$. Note that, since $\mathcal{Y}$ is binary, maximizing the probability for predicting one class can be equivalently done by minimizing the probability for predicting the other class. Hence, we may use Algorithm 2 to find the adversaries for both the positive and negative examples. Below we use the prediction probability for the class 1 as the score. Denote the score of $x$ and $x^r$ as $s(x)$ and $s(x^r)$. For $x \in A$, we know that $s(x) \geq s(x^r)$, and for $x \in B$, $s(x) \leq s(x^r)$.

If we are only allowed to perturb $k < n + m$ data points, to minimize AUC, we aim to solve the following program:

$$
\begin{aligned}
\text{minimize} \quad & \sum_{i=1}^{n}\sum_{j=1}^{m} \Big[ a_i b_j \hat{\mathbb{I}}(s(x_i^r), s(x_{j+n}^r)) + a_i(1 - b_j)\hat{\mathbb{I}}(s(x_i^r), s(x_{j+n})) \\
& + (1 - a_i)b_j \hat{\mathbb{I}}(s(x_i), s(x_{j+n}^r)) + (1 - a_i)(1 - b_j)\hat{\mathbb{I}}(s(x_i), s(x_{j+n})) \Big] \\
\text{subject to} \quad & \sum_{i \in [n]} a_i + \sum_{j \in [m]} b_j \le k, \\
& a_i \in \{0, 1\}, \quad i = 1, ..., n \\
& b_j \in \{0, 1\}, \quad j = 1, ..., m
\end{aligned}
$$

We may use standard mixed-integer programming solvers like Gurobi to solve the program. Here we use $a_i$ to denote whether data point $x_i \in A$ is perturbed, and $b_j$ to denote whether data point $x_{j+n} \in B$ is perturbed. The function $\hat{\mathbb{I}}(x, x')$ is an indicator function defined as

$$
\hat{\mathbb{I}}(x, x') \triangleq \begin{cases} 1, & \text{if } x > x', \\ 0.5 & \text{if } x = x', \\ 0, & \text{if } x < x'. \end{cases} \tag{25}
$$