[Reviews · NeurIPS 2019]

Reviewer 1



I thank the authors for a very well written paper with clear motivation, results and substantiation. Originality: This paper extends the work of Cohen et al. by considering smoothing distributions that can provide defenses to l0 attacks. This seems like a natural next step given the work of Cohen et al., The focus on structured classifiers (decision trees in this case) is novel. Decision trees have a natural structure which lends itself to tighter certificates for l0 attacks. The paper also provides some tricks to handle numerical issues when applying their approach to larger datasets like Imagenet. Quality: The paper is of very high quality and well written. The theorem statements and proofs seem accurate. (I haven't looked over the proof of Lemma 6, but from a cursory glance, looks right). However, on the empirical side, I have the following concerns. -- The baseline of using isotropic Gaussian smoothing (which is tight and suited for l2 robustness) seems too weak. There are some missing key comparison to previous work on l0 defenses, for example those based on sparse recovery (via Fourier transforms) considered in Bafna et al. 2018. -- Are the numbers reported in Table 1 and 2 performing a max over \alpha (the noise parameter?) If so, what's the corresponding clean accuracy? Ideally, one would report the certificates at all radii for each value of \alpha. The main concern with smoothing approach is that the noise needed to be added to get any meaningful robustness could be too large to allow good classification. Hence, it's important to see the clean accuracy numbers. Clarity: The paper is mostly well written and easy to follow. Here are some suggestions -- The theoretical results for l0 could be presented in a way that reflects how the noise parameter \alpha scales with dimension (assuming radius of l0 fixed). -- The relation between the warmup and the main distribution isn't very clear. What's the main intuition that carries over, and what changes? -- (Minor) H*-1 looks like inverse in Eq(2) Significance: The significance of the approach proposed for l0 defenses depends on the comparison to other existing approaches which hasn't been addressed in this work. The restriction to structured classes (say decision trees) is interesting, but upfront seems hard to extend to realistic settings due to a) Decision trees seem very tailored to l0. It's hard to think of similar structures for other lp norms b) Imposing structure might not allow certification of neural networks and other SOTA methods that would be most relevant. Questions to author: a) How many values of \alpha were considered in the experiments? b) From the warmup results, it seems that \gamma should grow exponentially with d for fixed R. This seems quite restrictive. -- Author feedback response -- I thank the authors for providing clarification on some questions. Regarding significance, unfortunately, I still feel the current presentation makes it difficult to see what are the key technical ideas (for certification for the discrete case) can be used in other distributions. Regarding decision trees, I agree it's an interesting observation but i still believe it's fairly specialized to the structure of decision trees and l0 norm. Therefore, I retain my overall score. Small clarification regarding comparison to other approaches: Since this is the first work to study "certified" defenses in l0 norm, apart from just the certification procedure, I think it's important to understand the tradeoffs introduced by the certified aspect. Hence, i recommended comparing to other defenses even if "heuristic/empirical". For example, the assumption in Bafna et al., is definitely not guaranteed to hold, but it would be illustrative for a reader to see how restrictive/helpful certification and different assumptions are in practice.

Reviewer 2



Update: I thank the authors for their clarifications, as I feel I can properly evaluate the paper now. I hope they make significant improvements to the paper to improve it's presentation, and consider presenting the ideas in a more straightforward, intuitive manner rather than hiding the core idea (which is actually quite simple, now that I understand properly) behind unnecessarily complex prose. R3's suggestions (and the rest of the reviewer feedback) are a good start, but also having a visual representation or a figure would also aid in making this easier to understand, so that the core idea of this work is able to reach a greater audience. ===================== Let me preface this review with the following: overall, after spending much effort on this paper, my experience is that the ideas were presented in a fairly difficult, unintuitive manner, and so I am not confident that I was able to fully understand everything correctly. As such, I have a couple questions for the authors that I hope they can clarify, after which I hope I can give the authors a proper reviewing score (the current score is not my final evaluation). 1) Lemma 1 just doesn't make intuitive sense to me. What is Equation (2) supposed to capture? I found section 3.1 to be overall rather unclear. 2) Equation 6 seems to imply that the L0 perturbation isn't really a true L0 perturbation, but one restricted to lie on some K-wide grid? 3) I understand from 4.3 that it is possible to turn a Gaussian perturbation into an L0 distance for binary vectors with a simple thresholding operator after applying Gaussian noise. However, the paper claims that this is a bijection, but I don't see any description on going back from an L0 distance to an L2 distance. Also, is there some proof for the claim that the composition perturbation is always tighter than the Gaussian perturbation? 4) In the experiments, is the comparison to the Gaussian perturbation somehow using the thresholding as described in 4.3? Or is it just naively applying the Gaussian perturbation and using it as a lower bound on the equivalent thresholded L0 perturbation? If it's the latter, then this is not really a useful competitor for comparison, as they are trying to certify against vastly different perturbation models, so it should be obvious that the approach designed for L0 perturbations does better when certifying L0 radii than the approach designed for L2 perturbations. The paper does mention that the L0 certificate is derived from the Gaussian perturbation, but exactly how this is done is not clear to me. It's not necessarily bad to include them for comparison, but it's a good idea to make it clear in the paper that these results are likely to be expected by design. 5) What is the value of K used in the experiments? How does it effect the certification abilities? 6) At the end of the last set of experiments in section 5.3, the authors mention that they compare the exact certificate vs the one derived from Lemma 1 and measure the average difference across the training data. But just before then they were evaluating the method with AUC. Can the authors provide a consistent comparison? Even better, include both Lemma 1 and exact certification results in the AUC curves in Figure 4?

Reviewer 3



== high-level overview == A recent series of papers culminating in [1] have shown that taking the ensemble prediction of any classifier over additive Gaussian corruptions of its input yields a new classifier that is certifiably robust in L2 norm. However, it has remained unclear whether other randomization schemes (besides additive Gaussian corruption) would yield natural robustness guarantees under other distance metrics (besides L2). This paper proposes a new randomization scheme which yields a natural robustness guarantee under the Hamming distance (a.k.a the L0 norm). Using this method, the authors obtain certified robustness guarantees in L0 norm on ImageNet (and binarized MNIST). I recommend accepting this paper. Originality: this paper is the first to give a randomization scheme for randomized smoothing which confers robustness in Hamming distance. Quality: the technical work is of high quality. Clarity: the writing is confusing in places, and I would not be surprised if some of the other reviewers have trouble understanding parts of the paper. Significance: the paper shows how to obtain neural network-based classifiers that are certifiably robust in Hamming distance, which is a significant result. == detailed overview (this may be helpful to the other reviewers) == Consider the problem of classifying strings "x" of characters from some discrete alphabet, and consider the following randomization scheme "\phi": for each letter x_i, with probability (1 - \alpha), replace that letter with a new character chosen uniformly at random from the alphabet, and with probability \alpha keep that letter in place. For example, if the alphabet is the English alphabet, and x is the string "quickbrownfox", and \alpha = 0.8, then here are five draws of the random variable \phi(x): qufckbvownfox, buiczbruwnfoo, qucckbqopnfox, qucckbrownfox, qzqckbgpwnfox. Ok, now let "f" be a base classifier which maps from strings to classes, and define the corresponding smoothed classifier "g" as g(x) := argmax_{class c} Pr[ f( \phi(x) ) = c ], i.e. g(x) returns the most probable prediction by f of the random variable \phi(x). This paper proves that g is certifiably robust under the Hamming distance. That is, if we measure p := Pr[ f( \phi(x) ) = y ], where y := g(x) is the top class, then we can certify that for any input x' within some L0 ball around x, g(x') = y. The paper applies this guarantee to the domain of image classification, where the "alphabet" consists of the 256 integers 0, 1, 2, ..., 255, and each ImageNet image can be viewed as a "string" of length 224 x 224 x 3. That is, to sample from \phi(x), you consider each pixel x_i, and w.p. (1 - \alpha) you swap out that pixel value for another pixel value chosen uniformly at random from the range 0, 1,2, ..., 255. You can see examples of these corruptions in Figure 5 in the appendix. Ok, here's how you prove the robustness guarantee. Let x be the original input and let x' be some hypothetical perturbed version. To prove that g(x') = y, it suffices to prove that Pr [ f ( \phi(x') ) = y ] > 0.5. However, all we know about f is p :=Pr [ f ( \phi(x) ) = y ]. This suggests the following optimization problem over functions: what is the minimal value of Pr [ f ( \phi(x') ) = y ] subject to the constraint that Pr [ f ( \phi(x) ) = y ] = p? In the Gaussian case studied in [1], the function f* which attains this minimum is the function f* which returns y on a superlevel set of the likelihood ratio function, i.e. a set of the form {z: \phi(z | x) / \phi(z | x') >= c } for some c chosen so that P[ f* ( \phi(x) ) = y] = p]. (I am using \phi(z | x) to denote the probability of z under the random variable \phi(x).) In the discrete case studied here, however, it is impossible to choose some c for which P[ f* ( \phi(x) ) = y] = p] is exactly equal to c. Therefore, the optimal f* is actually a *randomized* classifier which *always* returns y on a set of the form {z: \phi(z | x) / \phi(z | x') > c } and *sometimes* returns y on a set of the form {z: \phi(z | x) / \phi(z | x') = c }, where the probability in the "sometimes" is carefully tuned to ensure that P[ f* ( \phi(x) ) = y] = p. This analogous to how the Neyman-Pearson UMP test is sometimes a randomized test. Under the randomization scheme \phi(x) proposed in this paper, the likelihood ratio function \phi(z | x) / \phi(z | x') is a function of just ||z - x||_0 and ||z - x'||_0, i.e. the Hamming distances between z and x, and between z and x'. However, to determine what the threshold "c" should be, you need to face the hairy question of computing the size of each likelihood ratio region, i.e the cardinality of the set of images/strings that are hamming distance "u" away from x, and hamming distance "v" away from x'. Sadly, there is apparently no way to do this computation in closed form; it needs to be done on a computer. This takes four days (!!) for ImageNet. But the silver lining is that by symmetry, you don't need to do this separately for each image pair (x, x'), you just need to do it once for each Hamming radius (Lemma 5). The experiments were thorough. The authors made sure to compare against relevant baselines, e.g. using Gaussian randomized smoothing to certify a Hamming ball. == feedback to the authors == -- The main contribution of this paper is the Hamming distance guarantee in section 4.2, and the associated algorithms / experiments. I think the submission currently under-emphasizes this contribution, and over-emphasizes the significance of both Lemma 1, and the uniform distribution analysis in section 4.1. Overall, I would recommend re-working the paper so as to put the Hamming robustness guarantee front-and-center. Why do I say that section 4.1 is over-emphasized? Because the L1 and L-inf guarantees that one can derive for the uniform distribution are very bad. (That's presumably why this section is pitched as a "warm-up.") For example, on 224x224x3 ImageNet, with \gamma = 0.5 (which is already a huge amount of noise), if the probability of the top class is p=0.99, then your bound in Proposition 4 will certify a L-infinity radius of only 0.001139 / 255, which is tiny. So I'm honestly not sure if it's even worth putting this in the main paper. Why do I say that Lemma 1 is over-emphasized? Because (a) Lemma 1 is a straightforward extension of the analysis from [1], and (b) I'm not sure if there are any good applications of Lemma 1 besides the Hamming distance guarantee. That is: I assume that the reason why you factor out Lemma 1 into a separate statement from the Hamming guarantee is so that others can build on it independently, but I'm just not sure if it's possible to do so. Additionally, I think that it makes your paper needlessly confusing to first present the abstract theorem first, and to then later instantiate that theorem in the form of the Hamming guarantee. I recommend the reverse order: you should first present the Hamming guarantee, and *then* note that the analysis can be generalized into Lemma 1 (which could live in the appendix). -- I think you should emphasize that the Hamming guarantee is *tight* w.r.t all measurable classifiers (just like the Gaussian / L2 guarantee from [1]). That is: any perturbation outside the certified Hamming radius is adversarial in the worst case over the base classifier. This suggests that your randomization scheme is naturally suited for Hamming distance robustness. -- Could you publicly release \rho^{-1}_r (0.5) for standard datasets like ImageNet so that others do not have to run the 4-day computation? -- I am ambivalent about the decision tree certificates in Section 4.2. On the one hand, it nicely demonstrates that making assumptions on the base classifier beyond just "p", one can obtain better certificates. On the other hand: (1) The main selling point of randomized smoothing is that it applies to large neural nets. There are perhaps better ways than randomized smoothing to certify the robustness of decision tree-based classifiers. (2) The assumption that the tree can only use each feature once seems restrictive? (3) Since both the smoothed classifier's prediction and the worst-case adversarial perturbation can be computed exactly using dynamic programming, I have a suspicion that a smoothed decision tree can be equivalently re-interpreted as a different kind of certifiably robust classifier altogether. (Though that isn't necessarily a _bad_ thing.) -- I recommend deleting Remark 2. The reference [35], which assumes countability, is 69 years old. Today, the extension of the NP lemma to randomized tests in both countable or uncountable domains is extremely common knowledge: see http://www.math.mcgill.ca/dstephens/OldCourses/557-2008/Handouts/Math557-07-HypTesting-Examples.pdf, or https://ocw.mit.edu/courses/mathematics/18-443-statistics-for-applications-fall-2003/lecture-notes/lec20.pdf for example. -- I think you should make clearer that the four-day computation will precompute \rho for *every* ImageNet image. You don't need to do the four-day computation separately for each image. I was confused about this for a while. -- The legend of Figure 3(a) makes it seem like the number of distinct likelihood ratio regions depends on the dataset, whereas in reality it just depends on the dimension of the dataset. -- Throughout the paper, you call your noise distribution / randomization scheme a "perturbation." I recommend using the term "noise distribution" or "randomization scheme" instead. Usually, in the adversarial robustness literature, a "perturbation" refers to a specific adversarially-chosen perturbation vector. -- Throughout the paper, you call a specific perturbation vector an "adversary." Usually, in the adversarial robustness literature, an adversary is an algorithm for generating a perturbation vector (like PGD, or FGSM, or Carlini-Wagner), rather than the adversarial perturbation itself. -- The caption to Table 1 states that "the first two rows refer to the same model with certificates computed via different methods." Just to be clear, it's the same *base classifier* in both rows, but the model whose robustness you are actually certifying (i.e. the smoothed classifier) is different between the two rows. -- The use of the L1 norm in Lemma 5 is confusing. [1] https://arxiv.org/abs/1902.02918

[Author Response · NeurIPS 2019]

**General responses**: We thank all the reviewers for helpful insights, comments, and suggestions.

• (R1 & R2) Our $\ell_0$ certificate and baseline: providing robustness certificates in discrete spaces is highly non-trivial.

Brute-force solution would require checking $\binom{d}{r}(K+1)^r$ prediction values. To our best knowledge, this work is the

first one to establish an actionable $\ell_0$ robustness certificate besides adapting certificates in other norms to $\ell_0$ norm. In

addition, our certificate is *tight* and *applicable* to *any* measurable classifiers, and we theoretically and empirically

demonstrate its feasibility to large problems. Under this circumstance, we believe the most appropriate (strongest)

baseline is the adaptation of the state-of-the-art certificate in other norms ($\ell_2$ via the Gaussian perturbations).

• (R1 & R3) Extra assumptions / decision trees: the significance is in the elaboration that adding an extra assumption

*always* leads to a *stronger* certificate (§4.3-4), which was not observed in prior work. An extra assumption is not

difficult to find in deep networks, e.g., overall $L$-Lipschitz continuity or a matrix characterization per layer. The

challenge is how to convert it to an actionable certificate, where we hope our finding would enable and solicit further

investigations. The decision tree example also has moderate significance due to its unique role in interpretability.

• (R1 & R2) The warmup example: it is only used for warming up how to use the theoretical foundation (§3.1-2).

Despite the tightness, the distribution is restrictive (supp$(\phi) \neq \mathcal{X}$), and thus the certificate is limited even if $p = 1$.

**Review 1**: We thank the reviewer for helpful comments, but hope the reviewer reconsiders the significance of the paper.

• (Contributions-1) The construction of the smoothing distribution: similarly to Cohen et al., our contribution is not in

the smoothing distribution but rather in deriving the certificate and associated algorithms.

• (Contributions-1) Suitable comparison: besides the general clarification, here we use Bafna et al. as an example to

clarify. They consider a *specific setting* where the clean input $x$ is approximately $k$-sparse in the Fourier domain.

Under this assumption, they used compressed sensing technique to get $x'$ from some corrupted input $x + \delta$ such that

$\|x - x'\|_2$ is small. Note that this is not a *robustness guarantee*, which further requires $f(x) = f(x')$. Empirically,

they only *show* successful defense against *some* adversarial attacks w.r.t. the base classifier (instead of w.r.t. the

recovery algorithm + base classifier), while we *guarantee* the robustness against *all* adversaries up to some radius.

• (Contributions-1) Speed of certification: for a perturbation $\phi$ on $\mathcal{X}$, it only requires *pre-computing* $\rho_r^{-1}(0.5)$ *once* for

each $r$, and then the certification for any base classifiers $f$ and data $x \in \mathcal{X}$ is done by checking if $p > \rho_r^{-1}(0.5)$.

• Contributions-3: generalization beyond Cohen et al. in uncountable $\mathcal{X}$ and randomized predictions is indeed not the

main contribution of our paper. (cf. general response #1 & 2)

• Accuracy: both Table 1 (MNIST) and Figure 4 (Bace) show the certified results for $\alpha = 0.8$ (tuned by validation).

The certified accuracy for different $\alpha$ and radii are available for ImageNet in Figure 3c. The raw accuracy / AUC can

be found in Figure 3c (ImageNet) and Figure 4 (Bace) when the x-axis is 0. The raw accuracy for MNIST is 0.983.

• Tuning $\alpha$: for MNIST, ImageNet, and Bace, we consider $\alpha$ values in $\{0.72, 0.76, \ldots, 0.96\}$, $\{0.1, 0.2, \ldots, 0.5\}$,

and $\{0.75, 0.8, \ldots, 0.9\}$, respectively. The information is available in Appendix C.1-3.

• How $\alpha$ scales with $d$: unfortunately the interaction between $\alpha$ and $d$ has no known closed form.

**Review 2**: We will try to make the paper more understandable. The responses are ordered according to the questions.

1. Lemma 1 solves the minimization problem in Eq. 1. It shows how much the probability of the correct class at $x$ can

decrease if we move to $\bar{x}$ for any (worst case) smoothed predictor provided that we know the likelihood ratio regions

for the perturbation distribution. The worst case classifier assigns one class for regions with high likelihood ratio,

and another class for the remaining regions. This is why they are ordered. The classification in the borderline regions

may need to be randomized (the middle case in Eq. 2).

2. We study $\ell_0$ robustness in discrete domains ($\ell_0$ distance in $\mathbb{R}^d$ would not work for any guarantees).

3. Bijection holds in $\{0, 1\}^d$, not in $\mathbb{R}^d$. We can map any Gaussian perturbation result into a discrete domain by

embedding $\{0, 1\}^d$ as points in $\mathbb{R}^d$, obtaining the continuous certified radius, and mapping back to an $\ell_0$ radius over

the binary points. Or we can map the Gaussian perturbation first to a discrete perturbation to obtain the guarantee.

In the latter case, the guarantee derived from the discrete perturbation is tighter than the one from the Gaussian

perturbation, since the more precise assumption yields a tighter result.

4. The $2^{\text{nd}}$ row in Table 1 is about the thresholding discussion. The $3^{\text{rd}}$ row in Table 1 and $\mathcal{N}$ in Table 2 adapts the $\ell_2$

certificate of radius $r$ using the Gaussian perturbations to an $\ell_0$ certificate of radius $\lfloor r^2 \rfloor$.

5. $K = 1$ (binary) in binarized MNIST and Bace, and $K = 255$ (pixel) in ImageNet. The certificate is always tight

regardless of $K$, and the complexity (pre-computing $\rho_r^{-1}(0.5)$) is $\Theta(d^3)$ (line 166), which does not depend on $K$.

6. Yes, we can provide additional comparisons in terms of AUC in the final version. However, such comparison will

only indirectly show the tightness of the bounds (the purpose of this experiment).

**Review 3**: We thank the reviewer for accurate explanations and helpful suggestions. We only clarify a few things.

• We will publicly release $\rho_r^{-1}(0.5)$, with the implementation, for all the $\alpha$ and $r$ used for MNIST and ImageNet.

• Despite the restrictive assumption, the smoothed tree yields large improvements even in the raw AUC (x=0 in Fig. 4).

• The first 2 rows in Table 1 refer to the same smoothed classifier; the discrete perturbation in $\{0, 1\}^d$ can be interpreted

as a Gaussian perturbation with denoising $\zeta$, so it can be certified in both ways. (also see the $3^{\text{rd}}$ response to R2)

[Meta-Review · NeurIPS 2019]

This paper extends the smoothing work by Cohen et al. by considering smoothing distributions that can provide defenses to l0 attacks. The reviewers and the AC found this to be a novel and interesting contribution.